# Hierarchical Multi-Grained Reasoning for Object Concept Learning

## Abstract

Human beings can easily understand object concepts involving attributes and affordances. Recently, to simulate this ability, Object Concept Learning (OCL) has been introduced as a new task to recognize attributes and affordances related to a given object. OCL is essentially a many-to-many mapping problem: While an object may possess multiple different concepts, a concept can also belong to multiple different objects. In this regard, the prevailing method of learning discriminative representation—which is effective in the single-mapping cases—often fails in OCL. Inspired by the reasoning mechanism of human beings, in this paper, we propose Hierarchical Multi-Grained Reasoning (HGR) for OCL, aiming to infer object-related concepts from coarse-to-fine and counterfactual grains. Specifically, we first propose a coarse-to-fine hierarchical reasoning module that exploits multi-step learnable prompts to progressively localize object-relevant concept information. Subsequently, multiple counterfactual samples are selected to strengthen the relations between objects and concepts, which further improves the reasoning performance. In the experiments, our method is evaluated on multiple benchmarks. Significant performance gains and extensive visualization analysis demonstrate the superiorities of our method.

## 1 Introduction

With the development of deep neural networks, many challenging tasks, e.g., object classification (Krizhevsky et al., 2012), detection (Ren et al., 2015), and segmentation (Huynh et al., 2021), have achieved many progresses. Most existing methods (Hameed & Khalaf, 2024; Gonthina & Prasad, 2024) often leverage the specific neural network to extract discriminative representation and construct accurate mappings between representations and corresponding categories, which is easily affected by environment variances. Instead, human beings could accurately understand object concepts involving attributes and affordances, which improves the performance and robustness of identifying objects. To imitate this ability, a task of object concept learning (Li et al., 2023b) is recently proposed, whose goal is to recognize the attributes and affordances related to a given object. Addressing this task is beneficial for promoting the development of embodied AI.

Towards this task, one straightforward solution is to follow traditional object classification to mine discriminative representations corresponding to object-related attributes and affordances, which further construct a one-to-one mapping between objects and attributes (or affordances). For example, the work Li et al. (2023b) designs a specific debiasing mechanism for learning discriminative object-agnostic attribute representations. However, in practice, an object could have multiple different attributes and affordances. Meanwhile, an attribute and affordance could also belong to multiple different objects. Taking Fig. 1 (a) as an example, a cake includes 'round', 'fresh', etc. And Pizza and Bowl all contain the 'round' attribute. Thus, for OCL, how to construct an accurate many-to-many mapping between objects and concepts is a critical challenge.

Of course, enhancing the discrimination of the learned object representation is still beneficial for improving the performance of object-to-concept mapping (Luo et al., 2023b; Almahairi et al., 2018). However, since many-to-many mapping is full of much uncertainty, only learning discriminative representation is easily affected by environment variances, which may weaken its performance and robustness. Therefore, simply learning discriminative representations is not sufficient for this task (Li et al., 2023b). However, based on current observations, humans could leverage the reasoning mech-

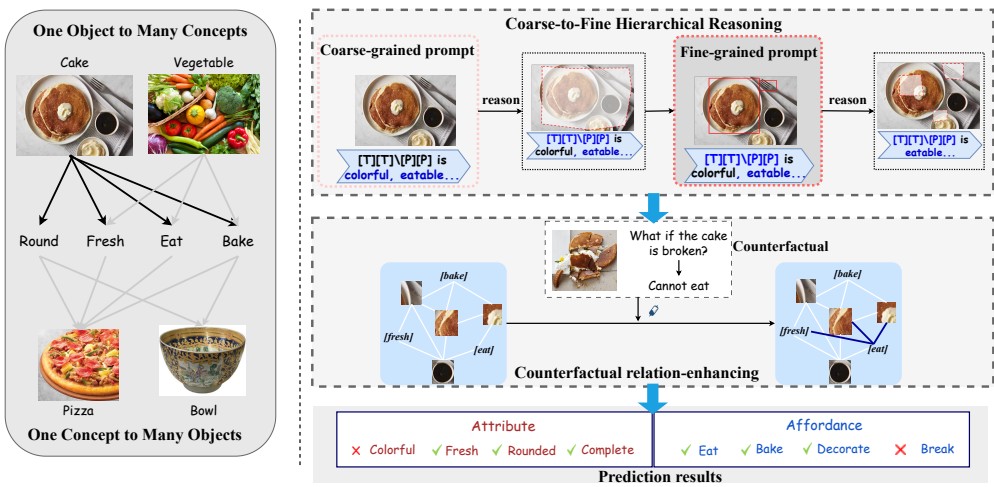

**(a) Object-to-Concept Learning**  **(b) Coarse-to-Fine Hierarchical Reasoning with Counterfactuals**

Figure 1: Compared with classical computer vision problems, Object Concept Learning is more challenging as it belongs to a many-to-many mapping problem. To this end, we explore designing a proper reasoning mechanism and propose a method of hierarchical multi-grained reasoning. We first exploit a series of learnable coarse and fine prompts to progressively focus on concept-relevant object information. Then, multiple counterfactual samples are selected to strengthen the relations between objects and concepts, which improves the accuracy of the learned object concepts.

anism to deepen their understanding of object concepts. To this end, in this paper, we first explore designing a dedicated reasoning method for learning object concepts.

Specifically, a method of Hierarchical Multi-Grained Reasoning (HGR) is proposed, consisting of a coarse-to-fine hierarchical reasoning module and a counterfactual relation-enhancing module. As illustrated in Figure 1, given an input image, we first design a series of coarse-grained prompts to promote the model to capture plentiful concept-relevant object information. On this basis, a series of dedicated fine-grained prompts are defined to accurately localize concept regions. Subsequently, to further strengthen the relations between concepts and objects, a graph neural network is designed to leverage multiple counterfactual samples to improve the performance of identifying concepts. Extensive experimental results and visualization analysis demonstrate the effectiveness of our method.

The contributions are summarized as follows:

(1) We first summarize OCL as a many-to-many mapping problem. To this end, how to construct an accurate many-to-many mapping between objects and concepts is a critical challenge. Meanwhile, a proper reasoning mechanism is designed to improve the reasoning accuracy.

(2) We propose a new reasoning method, i.e., Hierarchical Multi-Grained Reasoning, which integrates contextual information into coarse-to-fine reasoning process to more effectively identify the attributes and affordances of objects.

(3) Furthermore, due to the causal relationships between certain attributes and affordances, we design the counterfactual relation-enhancing model to accurately capture these causalities during training and improve recognition performance.

(4) Extensive experimental results and visualization analyses demonstrate the effectiveness of our method. Particularly, compared with state-of-the-art method (Li et al., 2023b), our method is **8.1%** and **3.9%** higher on attribute and affordance predictions.

## 2 RELATED WORK

**Attribute and Affordance Recognition.** Attributes recognition shares common background with other popular topics in research such as object detection (Ren et al., 2015), image segmentation (Huynh et al., 2021) and classification (Srinivas et al., 2021). It usually plays the role of

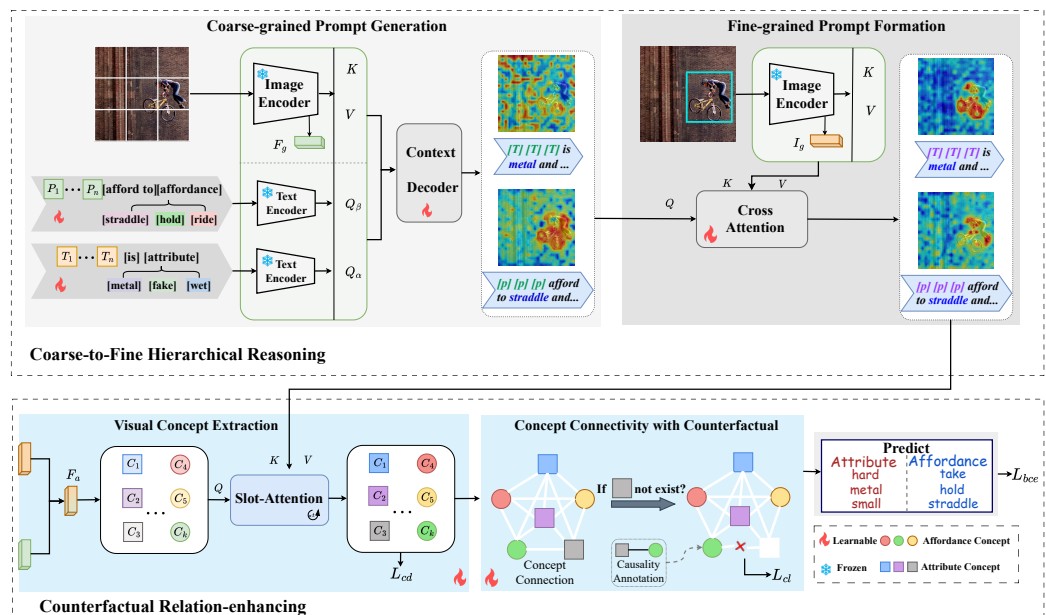

Figure 2: The details of Hierarchical Multi-Grained Reasoning (HGR). This method mainly consists of two components: Coarse-to-Fine Hierarchical Reasoning (CHR) and Counterfactual Relation-enhancing (CRE). Concretely, CHR first extracts visual tokens from the global image and generate coarse-grained attribute and affordance prompts. Subsequently, CHR refines the prompts by combining coarse-grained text prompts with localized visual information. The heatmaps illustrate the learning process from coarse to fine. Finally, we design a CRE to build the accurate relation between objects and concepts, which improves the performance of concept prediction.

mediator between pixels and higher-level concepts. However, visual attribute recognition has its unique challenges that set it apart from other visual tasks. The examples of these challenges include the large number of attributes need to be predicted and there exists many-to-many mapping rules between attributes and categories. Consequently, for attribute recognition, besides classifying the attribute directly, other methods incorporate both global and local information (Hwang et al., 2011) or excavate the intrinsic properties (Li et al., 2020) to enhance the performance.

Affordance recognition (Chen et al., 2023), aiming to reason about the objects' affordances in a scene through the input, leads to multivariant application in scene understanding Aarthi & Chitrakala (2017), human-object interaction (Antoun & Asmar, 2023) and so on. Most traditional affordance recognition methods rely on a Bayesian network (Friedman et al., 1997) or Support Vector Machine (Noble, 2006) to encode the dependencies between the object's global features and the affordances characteristics (Montesano et al., 2008; Uğur & Şahin, 2010). Deep learning-based methods learn the information of different modalities as prior knowledge and compensate them with the traditional methods to improve accuracy (Chen et al., 2023). For instance, Pinto et al. (Pinto & Gupta, 2016) utilized the multi-stage learning approach to collect affordances. Dadure et al. (Dadure et al., 2023) discusses knowledge representation and reasoning the target object itself. However, these methods often focus solely on affordance recognition. In practical scenarios, people often infer affordance based on observed attributes. For example, if we need to drink water but do not have a cup, we may find another hollow, hard object to hold the water. This reflects the importance of perceiving the relationship between attributes and affordances.

**Multimodal Prompting Methods.** Recently, many works (Lester et al., 2021; Liu et al., 2023; Alayrac et al., 2022) focus on prompting large pre-trained vision-language models to adapt to specific downstream tasks. The key idea of prompt engineering is to provide hints and other textual information to guide the pre-trained model in leveraging its existing knowledge to solve new tasks. The hints can be in the form of continuous vector representations, referred to as prompt tuning (Lester et al., 2021). This approach directly optimizes prompts within the embedding space of the model. The related work, such as (Dong et al., 2022), uses prompt tuning to improve the adaptation of pre-trained Vision Transformers to image and video understanding tasks. Additionally, CoOp (Zhou

et al., 2022b) introduces prompt tuning for visual tasks. They achieve this by converting context words into a set of learnable vectors to adapt them to the pre-trained vision-language model. Co-CoOp (Zhou et al., 2022a) further transforms static prompts into dynamic prompts to better handle category shifts. Chain-of-Thought Prompting (Wei et al., 2022) is a method to prompt the model by adding a series of intermediate reasoning steps. Each prompt in the chain incorporates contextual information, enabling the model to generate more coherent and contextually appropriate responses. Our work draws on the idea of the multi-step reasoning. In the object concept learning task, we explore the potential of large pre-trained models, which utilize prompt as a bridge between the image and visual concept for reasoning.

## 3 HIERARCHICAL MULTI-GRAINED REASONING

Figure 2 shows the framework of HGR model. Our work focuses on how to enhance the model's reasoning ability to understand object concepts. In this section, we present our method including Coarse-to-Fine Hierarchical Reasoning (Sec 3.1) and Counterfactual Relation-enhancing (Sec 3.2).

### 3.1 COARSE-TO-FINE HIERARCHICAL REASONING

Compared with concrete object categories, e.g., cake, object concept is much more abstract and involves plentiful information. To deepen the understanding of object concepts, we explore imitating human beings to perform coarse-to-fine reasoning to progressively localize concept-relevant object content, which improves the OCL performance.

#### 3.1.1 COARSE-GRAINED PROMPT GENERATION

The goal of Coarse-grained Prompt Generation is to create continuous vector representations as input prompts (see *Category-agnostic Prompting*) and obtain a more accurate attribute and affordance description including global visual contexts (see *Contextual Prompting*), which is helpful for a thorough understanding of visual content.

**Category-agnostic Prompting.** To adapt the large pre-trained vision-language model to the downstream recognition tasks, a common way is to use text prompt templates in CLIP, like "a photo of a [cls]", which primarily focuses on category semantics. Nevertheless, utilizing such hard text prompt templates presents challenges in generating generic attribute and affordance textual embeddings. This is because the original pretraining of CLIP focused on aligning with categorical semantics rather than high-level attributes and affordances concepts of images. To overcome this limitation, we aim to construct a set of learnable text prompts incorporating the prior knowledge of concepts. Recently works (Hassan & Dharmaratne, 2016; Li et al., 2023b) reveal that the attributes and affordances are shared between objects. Consequently, we construct a category-agnostic model and optimize prompts focusing on aligning with attribute and affordance semantics. We employ the prompt tuning (Lester et al., 2021) to construct a set of learnable text prompts $h$ incorporating the knowledge of attributes and affordances as:

$$
\begin{aligned}
h_\alpha &= [T_1]\,[T_2]\ldots[T_n]\,[\text{is}][\text{attribute}] \\
h_\beta &= [P_1]\,[P_2]\ldots[P_n]\,[\text{afford to}][\text{affordance}],
\end{aligned}
\tag{1}
$$

where $[T_i]$ and $[P_i](i \in 1,\ldots,n)$ are learnable token embeddings in attribute and affordance text prompt templates, respectively. This design ensures the category-agnostic text prompt template to learn the shared patterns of different categories.

**Contextual Prompting.** Since the nearby environment affects the recognition of attributes and affordances (Hassan & Dharmaratne, 2016), we devise a contextual prompt tuning approach that uses visual contexts to optimize the prompt features, making the generated textual embeddings capable of aligning visual content. Specifically, given an input image $X_i$, we extract the visual embedding $F_g \in \mathbb{R}^d$ from CLIP visual encoder $v(\cdot)$ and feature $F_M \in \mathbb{R}^{p \times d}$ from the $M$-th intermediate layer of CLIP visual encoder following (Zhou et al., 2023), where $p$ and $d$ separately denote the number of patches and feature dimension. And the input text $h_\alpha$ and $h_\beta$ are sent to the text encoder $f(\cdot)$, obtaining the text feature embeddings $H_\alpha \in \mathbb{R}^{N_\alpha \times d}$ and $H_\beta \in \mathbb{R}^{N_\beta \times d}$. $N_\alpha$ is the number of attributes and $N_\beta$ is the number of affordances. To encourage the text features to align with related visual

elements, we design the context decoder, where the features $F_M$ are used as the keys and values, and the text features $H_\alpha$ and $H_\beta$ are used as the queries:

$$\hat{H}_\alpha = \Theta_g(\Phi_g(H_\alpha), F_M) + H_\alpha, \hat{H}_\beta = \Theta_g(\Phi_g(H_\beta), F_M) + H_\beta, \tag{2}$$

where the $\Phi_g(\cdot)$ and the $\Theta_g(\cdot)$ represent the self-attention and cross-attention operation respectively. The self-attention mechanism allows for focusing on important contextual information for each word while reducing attention to irrelevant information. The cross-attention helps the model understand the semantic alignment between images and language, thereby providing a more accurate and meaningful joint representation. Through the residual connection "+", the language priors from the text features are preserved. Based on this, we can store the extracted text features $\hat{H}_\alpha$ and $\hat{H}_\beta$ with sufficient global visual information.

### 3.1.2 FINE-GRAINED PROMPT FORMATION

As our task is to identify the instance object's attribute and affordance, to further align the instance visual feature to attributes and affordances prompts, we employ ground-truth bounding boxes to crop the objects, and compute visual features $I_g \in \mathbb{R}^{1 \times d}$ through the CLIP visual encoder. Then, the text features $\hat{H}_\alpha$ and $\hat{H}_\beta$ are refined with the help of instance visual features $I_g$ by cross-attention $\Theta_I(\cdot)$:

$$\bar{H}_\alpha = \Theta_I(\hat{H}_\alpha, I_g), \bar{H}_\beta = \Theta_I(\hat{H}_\beta, I_g), \tag{3}$$

where $\bar{H}_\alpha \in \mathbb{R}^{N_\alpha \times d}$, $\bar{H}_\beta \in \mathbb{R}^{N_\beta \times d}$. From the above two steps, we first obtain category-agnostic text features with the global visual content, which helps to capture the attribute and affordance semantics comprehensively. Then, the fine-grained prompt formation is introduced to enable prompt features to concentrate on fine-grained visual contents, which guides the following visual concept reasoning.

### 3.2 COUNTERFACTUAL RELATION-ENHANCING BETWEEN OBJECTS AND CONCEPTS

For OCL, it is important to construct accurate connections between objects and concepts. To this end, we attempt to design multiple specific counterfactual samples to strengthen the object-concept relation, which further improves the reasoning accuracy.

### 3.2.1 PROMPT-GUIDED VISUAL CONCEPT EXTRACTION

We define a set of attribute concepts $C_\alpha = \left\{ c_i \in \mathbb{R}^D, i = 1, ..., k \right\}$ and affordance concepts $C_\beta = \left\{ c_i \in \mathbb{R}^D, i = 1, ..., k \right\}$, where $k$ denotes the number of concept and $D$ is the dimension of each concept. Each concept $c_i^{(t)}$ is initialized by visual feature $F_a$ and updated through attention and Gated Recurrent Unit (GRU) (Cho et al., 2014) operation over $t$ iterations, where $F_a$ is aggregated by the global image feature $F_g$ and instance image feature $I_g$. We project the $F_a$ and the text prompt features $\bar{H}_\alpha$, $\bar{H}_\beta$ dimension to $D$ by nonlinear transformations $Q$, $V$ and $K$ respectively. Dot-product is applied to generate an attention matrix $attn^{(t)}$:

$$attn_\alpha^{(t)} = \text{Softmax}(\frac{1}{\sqrt{d}}Q(C_\alpha^{(t)}) \cdot K(\bar{H}_\alpha)), attn_\beta^{(t)} = \text{Softmax}(\frac{1}{\sqrt{d}}Q(C_\beta^{(t)}) \cdot K(\bar{H}_\beta)), \tag{4}$$

where attention matrix $attn_\alpha \in \mathbb{R}^{k \times N_\alpha}$, $attn_\beta \in \mathbb{R}^{k \times N_\beta}$. To aggregate the input values $V$ to their assigned concepts, we use cross product operation and get the updates feature $U_\alpha^{(t)}$ and $U_\beta^{(t)}$:

$$U_\alpha^{(t)} = attn_\alpha^{(t)} \cdot V(\bar{H}_\alpha), U_\beta^{(t)} = attn_\beta^{(t)} \cdot V(\bar{H}_\beta), \tag{5}$$

where the aggregated updates feature $U_\alpha^{(t)}, U_\beta^{(t)} \in \mathbb{R}^{k \times D}$. The concept code $C_\alpha$ and $C_\beta$ are eventually updated with a GRU as $c_i^{(t)} = \text{GRU}\left(c_i^{(t-1)}, U^{(t)}\right)$, separately. In our experiment, the concepts are updated for $t = 3$ times.

### 3.2.2 CONCEPT CONNECTION NETWORK WITH COUNTERFACTUAL

**Concept Connection Network.** As is shown in the right part of Figure 2, the object rider afford to *ride* and *take* because the bicycle is *hard* and *metal*. Obviously, there are causal relationships

between some attributes and affordances. In order to enable the model not only recognizing these concepts but also learning the causal relationships between specific concepts, we design the relation-enhancing network. After obtaining the concepts $C_\alpha \in \mathbb{R}^{k \times D}$ and $C_\beta \in \mathbb{R}^{k \times D}$, we construct connection among concepts to reason out the specific attributes and affordances label.

Specifically, we seek to model an undirected attribute-affordance graph $G_a = \{V, \xi, \mathbf{A}\}$, where $\xi$ is the set of graph edges to learn and $\mathbf{A} \in \mathbb{R}^{k \times k}$ is the corresponding adjacency matrix. Each node $\nu \in V$ corresponds to one element of the visual concept $C_\alpha$ and $C_\beta$. And the size of $V$ is set to $2k$ . We define an adjacency matrix for the graph as $\mathbf{A} = \mathrm{softmax}_c \left( C_\alpha C_\beta^T \right) + I_d$, where $I_d$ indicates the identity matrix and softmaxc indicates we make softmax operation across the column direction.

$$M = \mathbf{A}C_\beta, \quad \widetilde{M} = \tanh\left( w_f^c * M + b_f^c \right), \tag{6}$$

where $w_f^c \in \mathbb{R}^{k \times k}$, $b_f^c \in \mathbb{R}^D$ indicate the trainable parameters. $\widetilde{M} \in \mathbb{R}^{k \times D}$ is the output of the concept connection network."*" indicates the multiplication operation. Each row of the affordance matrix $M$ represents a feature vector of a node, which is a weighted sum of the neighboring node features of the current node. Subsequently, we design a fusion operation to obtain the attribute and affordance classification feature. The fusion feature $\bar{F} \in \mathbb{R}^k$ is obtained by taking the dot-product of feature $F_a$ and matrix $\widetilde{M}$. The Softmax function $\phi(.)$ is used to generate a probability simplex over the $\bar{F}$, *i.e.*, $\phi(\bar{F}) = [p_i]_{i=1}^k$. Next, the affordance concept representation $F_\beta$ is derived by using a convex combination of the affordance features $\widetilde{M}$ weighted by their corresponding $p_i$, *i.e.*, $F_\beta = \sum_{i=1}^k p_i \cdot \widetilde{M}$. The attribute concept features $F_\alpha$ have the same fusion operation.

**Counterfactual Reasoning.** To better reason out attributes and affordances, we utilize the causality annotation from the benchmark to strengthen the connection among them. We add interventions on the attribute prompts by applying masks (Tang et al., 2020) to specific attribute elements and observing the corresponding affordances prediction results.

We formulate the masked attribute text prompt embedding as $H_{\alpha mask} = \bar{H}_\alpha * Mask$, where $Mask$ is generated following (Li et al., 2023b). Then, we sent the masked prompts to the visual concept extraction module and obtain the counterfactual affordance results $F_{\beta mask}$ from the concept connection network. Assuming there is a causal relationship between attribute $\alpha_i$ and affordance $\beta_i$. If there is a significant difference between the counterfactual affordance prediction result $\hat{y}_{\beta mask}$ and the original affordance result $\hat{y}_\beta$, it means that the model has learned the causal relationship between $\alpha_i$ and $\beta_i$. Conversely, it indicates that the model could not capture the causality. Based on this, we design the counterfactual loss as:

$$L_{cl} = \begin{cases} \max\{0, \gamma - (\hat{y}_\beta - \hat{y}_{\beta mask})\}, & \beta_i = 1, \\ \max\{0, \gamma + (\hat{y}_\beta - \hat{y}_{\beta mask})\}, & \beta_i = 0, \end{cases} \tag{7}$$

where $\gamma$ is a hyperparameter. We design two loss function $L_{cl}$ according to the different affordance label to promise the $L_{cl}$ should be a positive value.

Based on the above operation, we connect the attribute features $C_\alpha$ with the affordance features $C_\beta$. To promise the final predction results, we consider the following optimization strategy.

**Optimization.** The attribute and affordance features $F_\alpha$ and $F_\beta$ are sent to the different classifier, obtaining the predicted probability $\hat{y}_\alpha$ and $\hat{y}_\beta$ and calculating binary cross-entropy losses:

$$\mathcal{L}_{bce} = BCE\left(y_\alpha, \hat{y}_\alpha\right) + BCE\left(y_\beta, \hat{y}_\beta\right), \tag{8}$$

where $y_\alpha$ and $y_\beta$ are the attribute and affordance label. To capture different characteristics of images, different concepts should cover different visual regions. Therefore, each concept is enforced to keep it far from any other concept. We define a concept distinctiveness loss to achieve as:

$$\mathcal{L}_{cd} = \frac{1}{k(k-1)} \sum_{i,j}^k \frac{\langle U_i, U_j \rangle}{\|U_i\|_2^2 \|U_j\|_2^2}, \tag{9}$$

where $\| \cdot \|_2$ denotes L2-norm and $\langle \cdot, \cdot \rangle$ denotes the inner product operation. $U_i$ means the $i$-th updated concept feature, $U_j$ represents any other updated feature different from $U_i$. In this way, the concepts can capture different aspects of the image. The final loss function is $L_{total} = L_{bce} + \lambda_1 L_{cd} + \lambda_2 L_{cl}$. In the experiment, the $\lambda_1 = 0.1$ and $\lambda_2 = 1$.

Figure 3: The heatmaps of Coarse-to-Fine Hierarchical Reasoning. For each image, the top side indicates the heatmaps from the coarse-grained prompt generation module, and the bottom side indicates the heatmaps from the fine-grained prompt formation module.

Table 1: OCL accuracies (map). The baselines in the "N/A" fold means $\alpha$ and $\beta$ are calculated separately, without connection process. "$\alpha \rightarrow \beta$" means the $\beta$ is reasoned from the $\alpha$.

| Fold | Method | $\alpha$ | $\beta$ | $\mathcal{S}_{\text{ITE}}$ | $\mathcal{S}_{\alpha\text{-}\beta\text{-ITE}}$ | Fold | Method | $\alpha$ | $\beta$ | $\mathcal{S}_{\text{ITE}}$ | $\mathcal{S}_{\alpha\text{-}\beta\text{-ITE}}$ |
|------|--------|----------|---------|------|------|------|--------|----------|---------|------|------|
| | DM-V | 29.9 | 51.8 | - | - | | DM-$\alpha \rightarrow \beta$ | 28.8 | 52.4 | 15.5 | 14.0 |
| | HMa | 28.6 | 51.7 | - | - | | Attention | 23.9 | 49.0 | 17.8 | 15.5 |
| N/A | DM-att | 21.9 | 49.2 | - | - | $\alpha \rightarrow \beta$ | OCRN | 31.5 | 53.6 | 20.3 | 16.9 |
| | Vanilla CLIP | 23.6 | 49.6 | - | - | | Vanilla CLIP§ | 33.5 | 54.2 | 19.7 | 15.9 |
| | Vanilla CLIP† | **27.3** | **54.9** | - | - | | HGR | **39.6** | **57.5** | **20.9** | **17.4** |

## 4 EXPERIMENTS

We evaluate our method on the OCL datasets. We demonstrate that HGR improves attributes and affordances recognition performance and effectively enhances the causal effect between them. We also conduct experiments on Multi-task Indoor Scene Understanding and Weakly Supervised Affordance Grounding tasks to demonstrate that HGR can also perform well.

### 4.1 DATASETS AND BASELINES.

We consider three different tasks ranging from object concept learning, multi-task indoor scene understanding and weakly supervised affordance grounding. In object concept learning, we consider OCL (Li et al., 2023b) dataset, which is the first attribute-affordance reasoning dataset comprising 185,941 instances of 381 categories, 114 attributes, and 170 affordances. The SOTA competing methods include DM-V, DM-$\alpha \rightarrow \beta$ (Li et al., 2023b), HMa (Rumelhart et al., 1986), Attention (Vaswani et al., 2017), DM-att (Li et al., 2023b), OCRN (Li et al., 2023b), Vanilla CLIP (Radford et al., 2021). In multi-task indoor scene understanding task, we consider NYUd2 (Silberman et al., 2012) dataset including 1449 RGB-D images of indoor scenes with 40 object categories, 5 affordances and 11 attributes labels. The SOTA competing methods include PSPNet (Zhao et al., 2017), FastFCN (Wu et al., 2019), DeepLab V3 (Chen et al., 2017), VarReg (Shi et al., 2019) and Cerberus (Chen et al., 2022). In weakly supervised affordance grounding task, we consider AGD20K (Luo et al., 2022) dataset comprising of 20,061 exocentric images and 3,755 egocentric images, and is annotated with 36 affordances. The SOTA competing methods include Hotspots (Nagarajan et al., 2019), Cross-view-AG (Luo et al., 2022), Cross-view-AG+ (Luo et al., 2023a), AffCorrs (Hadjivelichkov et al., 2023), LOCATE (Li et al., 2023a).

### 4.2 THE PERFORMANCE OF OUR METHOD

Table 1 2 3 show the comparison of our HGR with the state-of-the-art models (Li et al., 2023a;b; Chen et al., 2022) on object concept learning, multi-task indoor scene understanding and weakly supervised affordance grounding benchmarks. Our approach consistently achieves superior performance compared to previous methods.

**Object Concept Learning.** According to the experiment in (Li et al., 2023b), we evaluate the affordance ($\beta$), attributes ($\alpha$), $\mathcal{S}_{\text{ITE}}$ and the $\mathcal{S}_{\alpha\text{-}\beta\text{-ITE}}$ performance. The mean Average Precision (mAP) is the evaluation metric for $\alpha$ and $\beta$. We follow the OCL and use $\mathcal{S}_{\text{ITE}}$ and $\mathcal{S}_{\alpha-\beta-\text{ITE}}$ as the

Table 2: Quantitative results on NYUd2 for Attribute, Affordance, and Semantic tasks.

| Attribute | | Affordance | | Semantic | | |
|---|---|---|---|---|---|---|
| Method | mIoU (%) | Method | mIoU (%) | Method | Input | mIoU (%) |
| PSPNet | 36.7 | PSPNet | 60.4 | FastFCN | RGB | 45.4 |
| DeepLab V3 | 38.1 | DeepLab V3 | 61.4 | VarReg | RGB | 50.7 |
| Cerberus | 45.3 | Cerberus | 66.3 | Cerberus | RGB | 50.4 |
| Cerberus+HGR | **46.0** | Cerberus+HGR | **67.2** | Cerberus+HGR | RGB | **50.6** |

Table 3: Comparison to state-of-the-arts from weakly supervised affordance grounding task on AGD20K dataset ($\uparrow$/$\downarrow$ means higher/lower is better).

| Method | Seen | | | Unseen | | |
|---|---|---|---|---|---|---|
| | KLD$\downarrow$ | SIM$\uparrow$ | NSS$\uparrow$ | KLD$\downarrow$ | SIM$\uparrow$ | NSS$\uparrow$ |
| Hotspots | 1.773 | 0.278 | 0.615 | 1.994 | 0.237 | 0.577 |
| Cross-view-AG | 1.538 | 0.334 | 0.927 | 1.787 | 0.285 | 0.829 |
| Cross-view-AG+ | 1.489 | 0.342 | 0.981 | 1.765 | 0.279 | 0.882 |
| AffCorrs | 1.407 | 0.359 | 1.026 | 1.618 | 0.348 | 1.021 |
| LOCATE | 1.226 | 0.401 | 1.177 | 1.405 | 0.372 | 1.157 |
| LOCATE+HGR | **1.193** | **0.432** | **1.233** | **1.331** | **0.379** | **1.210** |

causal relevant metrics. For fair comparison, we combine the Vanilla CLIP baseline method with the classifier, denoted as "Vanilla CLIP†" and "Vanilla CLIP§" in the N/A and $\alpha \to \beta$ fold respectively, to investigate the effect of our method.

As shown in Table 1, our approach outperforms the published state-of-the-art method (Li et al., 2023b) by **8.1** map on attribute and **3.9** map on affordance, respectively. Compared with the "Vanilla CLIP§", which concats the attributes and the affordances features together, our method improves the performance, attaining a 6.1 map enhancement on attribute and 3.3 map enhancement on affordance. These suggest that our method can effectively capture the object's attributes and affordances characteristics and own the ability to reason out the multi-label affordances from attributes. Furthermore, we observe that the performance of affordance ($\beta$) is better than attribute ($\alpha$). As mentioned in (Li et al., 2023b), the possible reason is that one object usually has various attributes and the attribute number is less than the affordance number.

In addition, the reasoning scores $\mathcal{S}_{\text{ITE}}$ and the $\mathcal{S}_{\alpha\text{-}\beta\text{-ITE}}$ which combines the recognition probability are higher than the baseline methods, which indicates that our method is benefit for reasoning the relationship between the attributes and affordances. In Figure 5, we show some visualization results. As is shown in these examples, compared with OCRN (Li et al., 2023b), our method not only predicts attributes and affordances more accurately but also correctly recognizes the causality pair, which further demonstrates the superiority of HGR.

**Multi-task Indoor Scene Understanding.** Multi-task indoor scene understanding is a task to parse attribute, affordance and semantic from a single image. The mean intersection over union (mIoU) score is the evaluation metric. To evaluate the scene understanding qualities and generalization ability of our proposed method, we add our method on the baseline method Cerberus (Chen et al., 2022). We set the number of attribute concepts $k_\alpha$ equals 6 and the number of affordance concepts $k_\beta$ equals 3. As shown in Table 2, HGR is added to the Cerberus and improves the performance significantly. Besides, in semantic parsing task, although the results do not perform well compared with VarReg, it still improves the Cerberus accuracy. These indicate that our HGR not only enhances the model's reasoning ability in the OCL benchmark but also helps to achieve joint inference in multi-task prediction. More details can be found in the appendix A.2.

**Weakly Supervised Affordance Grounding.** Since affordance understanding of interaction locations has garnered significant attention in the domains of robotics and computer vision, we conduct experiments on the weakly supervised affordance grounding (Li et al., 2023a) to evaluate the model's cognitive reasoning capabilities and generalization ability. Weakly supervised affordance grounding goal is to perform affordance grounding in the target object image where only the image-level labels are given without any per-pixel annotations. Kullback-Leibler Divergence (KLD), Similarity (SIM), and Normalized Scanpath Saliency (NSS) are used as metrics. As shown in Table 3, by designing the affordance prompt for incorporating the text knowledge during training, we report the accuracy

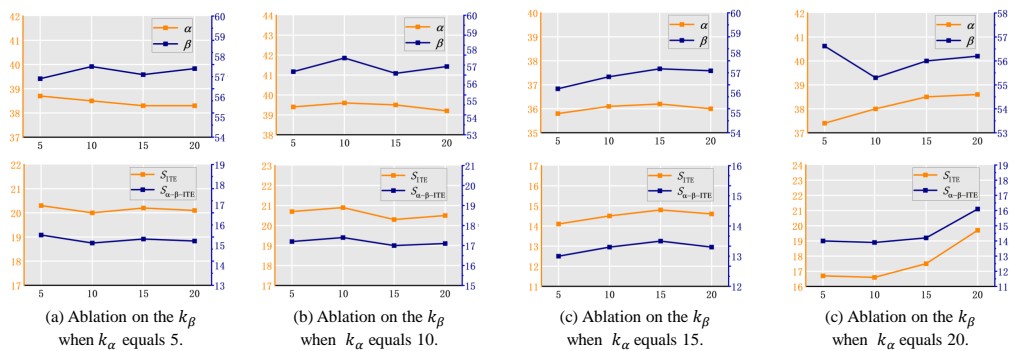

(a) Ablation on the $k_\beta$ when $k_\alpha$ equals 5.

(b) Ablation on the $k_\beta$ when $k_\alpha$ equals 10.

(c) Ablation on the $k_\beta$ when $k_\alpha$ equals 15.

(c) Ablation on the $k_\beta$ when $k_\alpha$ equals 20.

Figure 4: The ablation results of the concept number based on the OCL benchmark.

of our method combined with the baseline method LOCATE. Notably, our proposed method HGR makes an improvement over LOCATE. These results demonstrate the excellent adaptation ability of HGR. More details can be found in the appendix A.2.

### 4.3 ABLATION ANALYSIS OF EACH COMPONENTS

**Module ablation.** We validate the effectiveness of different high-level modules of our HGR, including Vanilla CLIP (Base), Coarse-to-Fine Hierarchical Reasoning (CHR), prompt-guided visual concept extraction (PVCE), and concept connection network with counterfactual (CCC). As shown in Table 4, each module contributes to the remarkable performance of HGR. CHR improves recognition performance through coarse-to-fine prompting learning. PVCE aggregates the learned fine-grained textual prompts into the visual space, enhancing the representation of visual concept features. Furthermore, CCC enhances the causal relationship between attributes and affordances through counterfactual reasoning, which promotes the accuracy of many-to-many mappings.

Table 4: Ablation study of the module reported on OCL benchmark.

| Method | $\alpha$ | $\beta$ | $\mathcal{S}_{\text{ITE}}$ | $\mathcal{S}_{\alpha\text{-}\beta\text{-ITE}}$ |
|--------|----------|---------|------------|--------------------|
| Base   | 33.5     | 54.2    | 19.7       | 15.9               |
| +CHR   | 37.8     | 55.8    | 19.7       | 16.1               |
| +PVCE  | 39.0     | 56.4    | 20.0       | 16.5               |
| +CCC   | **39.6** | **57.5**| **20.9**   | **17.4**           |

**Analysis of Contextual Prompt.** Table 5 presents the effects of prompt flow. We decompose the prompt reasoning into a two step refining process, where the first step is to generate prompt containing the global image information. The second row in Table 5 shows the results only by fusing the global image contextual with text prompts. The second step is to

Table 5: Ablation study of the different prompt step reported on OCL benchmark.

| Prompt | $\alpha$ | $\beta$ | $\mathcal{S}_{\text{ITE}}$ | $\mathcal{S}_{\alpha\text{-}\beta\text{-ITE}}$ |
|--------|----------|---------|------------|--------------------|
| global | 37.5     | 54.4    | 16.3       | 15.8               |
| local  | 24.8     | 40.6    | 11.5       | 9.3                |
| global+local | **39.6** | **57.5** | **20.9** | **17.4**        |

relay the prompt from the previous step and deepen the local instance corresponding prompts. The fourth row results indicate the performance of the second step, surpassing the performance of the global contextual prompt. However, only the instance-specific contextual prompt leads to poor performance, as shown in the third row. The results suggest that model can not align the attribute and affordance prompts with image features directly solely from local regions. The reason may be that most objects' interact with the nearby environment. Thus, the model is difficult to comprehend instance information without global image guidance. In addition, we report the hierarchical reasoning heatmaps in Figure 3. Our method could indeed focus on concept-related object regions progressively by means of coarse-to-fine prompts, which improves the accuracy of recognition concepts.

**Analysis of the Number of Concepts.** We study the impact on recognition results when the number of attributes $k_\alpha$ and affordances $k_\beta$ visual concepts differ. We report in Figure 4 the results by changing $k \in [1, 20]$ with a 5 interval. As can be observed in the OCL benchmark, when the values of $k_\alpha$ and $k_\beta$ are the same, the recognition accuracy is higher. This indicates that significant differences in visual concepts may lead to inaccurate mapping relationships. The recognition accuracy reaches its peak when $k$ equals 10. A reasonable explanation is that although OCL contains more attributes and affordances, the contents of images are more complex, learning too many concepts may increase the complexity of the network and thus influence the accuracy and stability of the model.

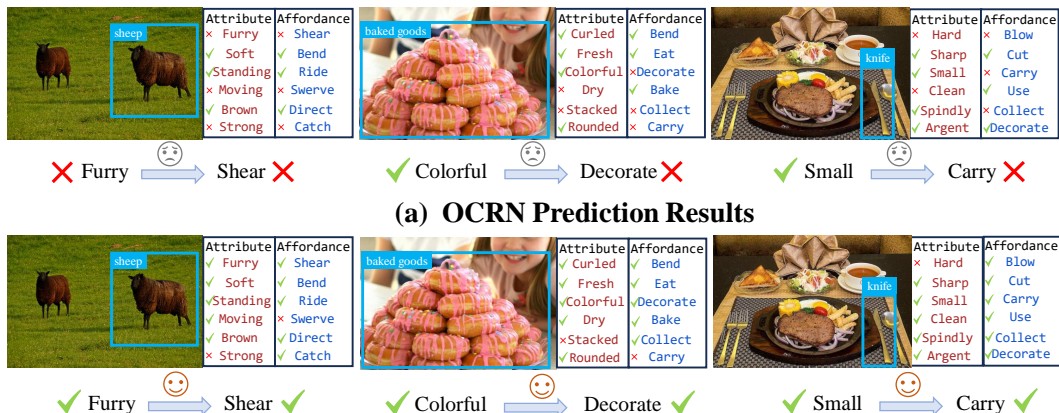

(a) **OCRN Prediction Results**

(b) **Our Method Prediction Results**

Figure 5: The ablation results of the baseline method (OCRN) and HGR. The attributes and affordances prediction results are shown in the image right part. The causal relation from dataset annotation is presented below each image.

## 5 CONCLUSIONS AND LIMINATIONS

For OCL, we explore introducing a reasoning mechanism to strengthen object concept learning. Concretely, we propose a Hierarchical Multi-Grained Reasoning (HGR) method, which consists of coarse-to-fine hierarchical reasoning module and counterfactual relation-enhancing module. Particularly, we first sent the entire image to the Coarse-to-Fine Hierarchical Reasoning module, obtaining the fine-grained prompt containing instance object concepts. Subsequently, multiple counterfactual samples are selected to strengthen the relations between objects and concepts, which further improves the reasoning performance. Experiment results show the effectiveness of HGR.

Notably, it still has much room for reasoning ability improvement. From the experiments, we found that although the CLIP improved the model's recognition of attributes and affordances, there is still much room for improvement. It is worth noting that capturing causal relationships between attributes and affordances requires deeper exploration. In the future, we plan to validate and optimize our method in a broader range of application scenes.

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

# A  APPENDIX

Table 6: Ablation of the $\gamma$ (equation 7) on OCL benchmark.

| $\gamma$ | $\alpha$ | $\beta$ | $\mathcal{S}_{\text{ITE}}$ | $\mathcal{S}_{\alpha\text{-}\beta\text{-ITE}}$ |
|---|---|---|---|---|
| 0.1 | 38.7 | 56.1 | 19.7 | 16.9 |
| 0.3 | **39.6** | **57.5** | **20.9** | **17.4** |
| 0.5 | 38.5 | 56.4 | 19.9 | 17.0 |
| 0.7 | 38.0 | 56.0 | 18.2 | 16.6 |
| 1.0 | 35.2 | 55.6 | 18.0 | 16.2 |

Table 7: Ablation of learnable prompts $n$ on OCL.

| $n$ | $\alpha$ | $\beta$ | $\mathcal{S}_{\text{ITE}}$ | $\mathcal{S}_{\alpha\text{-}\beta\text{-ITE}}$ |
|---|---|---|---|---|
| 10 | 38.9 | 57.0 | 20.1 | 17.0 |
| 12 | **39.6** | **57.5** | **20.9** | **17.4** |
| 14 | 39.2 | 57.1 | 20.5 | 17.2 |
| 16 | 38.7 | 56.8 | 20.2 | 16.9 |

Table 8: Ablation of concept connection network on OCL.

| Method | $\alpha$ | $\beta$ | $\mathcal{S}_{\text{ITE}}$ | $\mathcal{S}_{\alpha\text{-}\beta\text{-ITE}}$ |
|---|---|---|---|---|
| w/ Linear Network | 39.2 | 56.0 | 19.9 | 16.1 |
| w/o Linear Network | **39.6** | **57.5** | **20.9** | **17.4** |

For object concept learning, to mitigate the significant uncertainty of many-to-many mappings, we proposes HGR method, aiming to exploit the coarse-to-fine hierarchical reasoning module to perform object attributes and affordances recognition, and leveraging multiple counterfactual samples to strengthen the relations between objects and concepts. In the appendix, we provide implementation details, additional analyses, various ablation studies, and more visualization results.

## A.1  EXPERIMENTAL SETUP

**Implementation details.** We use the CLIP model VIT-L/14@336px as our backbone. The length of learnable attribute and affordance prompt embeddings $n$ is set to 12 and the CLIP visual encoder parameters are frozen. For OCL, the concept number $k$ in the best experiment results is 10 and the $\gamma$ is 0.3. The ablation experiments setting are based on the $k = 10$. The model learns with batch size 128 and SGD learning rate 1 for parameter optimization. For NYUd2 benchmark, the counterfactual reasoning cannot be used since there is no causality annotation. Thus, we add our concept extraction module to the baseline network. The experiments use the standard SGD optimizer with a learning rate of 7e-3, momentum 0.9, and batch size 2. For AGD20K, the names of the affordances corresponding to the image labels have been added to the prompt template. And we set the $k = 5$ and batch size 16. SGD with learning rate 1e-3, weight decay 5e-4 is used for parameter optimization. In addition, the metrics in OCL also include the $\mathcal{S}_{\text{ITE}}$ and $\mathcal{S}_{\alpha\text{-}\beta\text{-ITE}}$, which combine the actual affordance probability and counterfactual output following (Li et al., 2023b) to verify the performance of reasoning. All experiments are conducted in PyTorch-1.10 with two NVIDIA RTX A6000. More details can be found in the appendix.

## A.2  EXPERIMENTAL DETAILS

Our method can be considered an independent module that can be flexibly integrated into existing methods. For NYUd2 (Silberman et al., 2012), since the original method involves joint training of attributes and affordances, we can incorporate our method into the original approach by constructing learnable prompts using the labels of attributes and affordances. For AGD20K (Luo et al.,

2022), since the affordance labels are available, we build coarse-grained prompt learning on the exocentric branch and construct fine-grained prompt formation on the egocentric branch. Subsequently, the visual concept extraction module maps the prompts to discriminative visual features. The optimization includes the loss of our module as well as the loss of the original framework. Additionally,incorporating our module does not alter the optimization process of the original network.

## A.3 MORE ABLATION EXPERIMENTS OF HYPER-PARAMETERS AND MODULES

For our method, we utilize the hyper-parameter $n$ for the length of learnable prompts, the hyper-parameter $\gamma$ for the loss $L_{cl}$ (equation 7) and concept connection network to connect the attribute $\alpha$ and affordance $\beta$. Here, we take the OCL dataset to perform an ablation analysis of hyper-parameters and concept connection module. And we only change these hyper-parameters and keep other modules unchanged.

**Analysis of $\gamma$.** The hyper-parameter $\gamma$ in equation 7 is a threshold, which can be dynamically adjusted. From the Table 6, we find that when $\gamma$ is set to 0.3, the corresponding evaluation metrics get the best performance.

**Analysis of $n$.** From the Table 7, we can find that the performance initially improves with an increase in the value of n.However, within the range of lengths from 12 to 16, we notice a decline in performance, which suggests that excessively long learnable prompts could involve redundant information. Therefore, an appropriate value is $n = 12$.

**Analysis of concept connection network.** The concept connection network goal is to construct connection among concepts to reason out the specific attributes and affordances label. To validate the effectiveness of our designed network, we replaced matrix **A** in the network with a linear network for experimental analysis. The results in Table 8 emerges that replacing the adjacency matrix **A** with a linear network reduces performance. This indicates that our concept connection network is better at improving performance.

**Analysis of the Number of Concepts from the Complete Dataset.** We conduct experiments where the number of $k$ samples is equal to the total number of attributes (114) and affordances (170) in the entire dataset. The results are as shown in Table 9.

Table 9: More ablation of concept number $k$ on OCL. "att" is the abbreviation for attributes, and "aff" is the abbreviation for affordances. The values in "()" represent the number of $k$.

| $k$ | $\alpha$ | $\beta$ | $\mathcal{S}_{\text{ITE}}$ | $\mathcal{S}_{\alpha\text{-}\beta\text{-ITE}}$ |
|---|---|---|---|---|
| att(114)-aff(114) | 36.7 | 55.9 | 19.9 | 16.3 |
| att(114)-aff(170) | 35.9 | 55.1 | 19.5 | 15.9 |
| att(10)-aff(10) | 39.6 | 57.5 | 20.9 | 17.4 |
| Vanilla CLIP§ | 33.5 | 54.2 | 19.7 | 15.9 |

When the number of $k$ equals the number of attributes and affordances, the model's performance could be improved compared with the Vanilla CLIP baseline. However, the model's performance decreases compared with the "attr(10)-aff(10)". The experimental results reveal that an excessive number of $k$ samples decreases performance, and a significant difference between attributes and affordances also results in poor performance. Learning too many visual concepts will increase the network's complexity and introduce information redundancy, degrading performance.

Table 10: Ablation of GRU on OCL.

| Method | $\alpha$ | $\beta$ | $\mathcal{S}_{\text{ITE}}$ | $\mathcal{S}_{\alpha\text{-}\beta\text{-ITE}}$ |
|---|---|---|---|---|
| w/o GRU | 39.4 | 57.2 | 20.7 | 17.3 |
| w/ GRU | **39.6** | **57.5** | **20.9** | **17.4** |

**Analysis of the GRU and Concept Update.** From the Table 10, it can be observed that removing the GRU leads to a slight decrease in performance. This is because the GRU is used to refine the shape of the regions corresponding to the extracted concepts. Different attributes and affordances of an object correspond to different region. Therefore, it is necessary to provide a better regional boundary for these concepts.

Table 11: Ablation of Concept Update on OCL.

| Method | $\alpha$ | $\beta$ | $\mathcal{S}_{\text{ITE}}$ | $\mathcal{S}_{\alpha\text{-}\beta\text{-ITE}}$ |
|---|---|---|---|---|
| w/o concept update | 38.8 | 57.0 | 20.5 | 17.1 |
| w/ concept update | **39.6** | **57.5** | **20.9** | **17.4** |

The experimental results in Table 11 indicate that performance declines when concept update is not employed. This is because each concept feature is initialized by visual features and then progressively updated to different object regions through attention-based clustering. Without concept update, different concepts may intertwine, making it more difficult to complete the recognition task.

Table 12: Ablation of Ground-truth Bounding Boxes (bbox) on OCL.

| Method | $\alpha$ | $\beta$ | $\mathcal{S}_{\text{ITE}}$ | $\mathcal{S}_{\alpha\text{-}\beta\text{-ITE}}$ |
|---|---|---|---|---|
| OCRN w/ bbox | 31.5 | 53.6 | 20.3 | 16.9 |
| OCRN w/o bbox | 28.6 | 49.8 | 17.7 | 15.5 |
| HGR w/o bbox | **33.9** | **52.3** | 20.1 | **17.0** |

**Analysis of the Bounding Boxes.** We replace the bounding boxes with the predicted boxes extracted by Faster R-CNN. The results of our experiments are shown in Table 12. From the experimental results, it can be observed that there is a decline in performance after using the pre-trained Faster R-CNN to extract the prediction boxes. However, our method can still enhance the performance of the baseline.

## A.4 IMBALANCE LEARNING AND ZERO-SHOT CAUSAL LEARNING

### A.4.1 IMBALANCE LEARNING

There is an imbalance in the distribution of attributes and affordances for objects. The original distribution of attributes and affordances in the OCL dataset is also imbalanced. To illustrate the impact of the imbalance ratio on our method, we set the same imbalance ratio $r$ for both attributes and affordance, and we conduct experiments.

Table 13: More ablation of imbalance ratio $r$ on OCL. The values in "()" represent the imbalance ratio.

| $r$ | $\alpha$ | $\beta$ | $\mathcal{S}_{\text{ITE}}$ | $\mathcal{S}_{\alpha\text{-}\beta\text{-ITE}}$ |
|---|---|---|---|---|
| OCRN-r(100) | 26.4 | 49.8 | 15.5 | 14.3 |
| HGR-r(100) | **29.5** | **52.0** | **16.1** | **14.7** |
| OCRN-r(50) | 29.1 | 52.3 | 18.2 | 16.1 |
| HGR-r(50) | **34.2** | **55.1** | **18.9** | **16.6** |
| OCRN-r(10) | 32.4 | 54.1 | 20.4 | 16.7 |
| HGR-r(10) | **39.2** | **57.2** | **20.5** | **17.3** |

By setting the same imbalance ratio in the OCL baseline OCRN and our method HGR, it can be observed in Table 13 that our method still improves performance. This indicates that our approach can adapt to imbalance cases.

### A.4.2 ZERO-SHOT CAUSAL LEARNING

To further demonstrate the effectiveness of our method, we conducted experiments on the zero-shot causal setting, and the results are as shown in Table 14.

Table 14: Performance results for OCL dataset on zero-shot causal learning task.

| Method | $\alpha$ | $\beta$ | $\mathcal{S}_{\text{ITE}}$ | $\mathcal{S}_{\alpha\text{-}\beta\text{-ITE}}$ |
|--------|----------|---------|------------|-------------------------|
| OCRN | 30.0 | 52.5 | 16.3 | 14.1 |
| HGR | **37.5** | **56.1** | **17.3** | **15.2** |

The results indicate that our method can enhance the model's reasoning performance. In this setting, 300 attributes-affordances causality annotations are used as unseen causal relations. 785 attributes-affordances causality annotations are used as seen causal relations. For this setting, the concept number $k$ in the best experiment results is 10 and the $\gamma$ is 0.1.

## A.5 MORE EXPERIMENTS ON MEDICAL DOMAIN

To verify the effectiveness and generalizability of our method, we further conduct experiments on medical datasets. The results are shown in Table 15.

Table 15: Comparison of AUC scores with other state-of-the-art methods on fine-tuning classification task. The results are reported for ChestX-ray14 dataset.

| Method | Data portion 1% | Data portion 10% | Data portion 100% |
|--------|-----------------|-------------------|--------------------|
| GLoRIA | 0.6710 | 0.7642 | 0.8184 |
| BioViL | 0.6952 | 0.7527 | 0.8245 |
| MedKLP | 0.7721 | 0.7894 | 0.8323 |
| HGR | **0.7876** | **0.7963** | **0.8388** |

We have compared our method with the GLoRIA Huang et al. (2021), BioViL Boecking et al. (2022), and MedKLIP Wu et al. (2023) baselines. To ensure fairness, we follow the same protocol. The experiments are conducted on the ChestX-ray14 Wang et al. (2017) dataset. From the experimental results, it can be observed that our method can achieve better performance enhancement, which demonstrates that our method possesses generalization ability.

## A.6 FURTHER DISCUSSION

Although multi-label recognition is not a new problem, it is under explored in the Object Concept Learning (OCL) task. Multi-label recognition mainly focus on the mapping between images and categories. The main challenge of OCL lies in the many-to-many mapping relationships between objects and concepts. That is, an object could have multiple different attributes and affordances (concepts). Meanwhile, an attribute and affordance could also belong to multiple different objects. To overcome this, we propose a Hierarchical Multi-Grained Reasoning framework. For objects in an image, an attribute and affordance could belong to multiple different objects, which means concepts exist across objects. Thus, we first design object-agnostic prompts to enhance concepts-objects mapping precision. Subsequently, a mapping between object concepts is formed by integrating contextual global information and fine-grained local information. Furthermore, to make the mapping between objects and concepts more precise, we introduce counterfactual reasoning to identify causal relationships between certain attributes and affordances. This enhances the mapping connections between objects and these concepts, which in turn improves recognition performance. Significant performance improvements and extensive visual analysis have demonstrated the superiority of HGR.

### A.7 VISUALIZATION RESULTS

We test our approach on the Object Concept Learning (OCL) benchmark. The visualization results are as follows. Among them, Figure 7 is the baseline method, Figure 8 is our method, and Figure 6 is the coarse-grained and fine-grained prompt heatmaps. For Figure 7 and Figure 8, the right side of each image shows the predicted attribute and affordance of the object, and the bottom of the image shows the test results of causality. The experimental results further demonstrate that our approach not only achieves greater accuracy in attribute and affordance prediction, but also indicates that our method own the ability to understand causality. For Figure 6, the top side of each image show the heatmaps from the coarse-grained prompt generation module, and the down side show the heatmaps from the fine-grained prompt formation module. The results show that our method could capture the object-related attributes and affordances features in a coarse-to-fine manner. These results prove the applicability and superiority of our proposed method in complex scenarios, and lay a foundation for future object understanding application in a wider range of fields.

### A.8 VISUALIZATION RESULTS FOR CAUSAL RELATIONS

To further illustrate that our method HGR can learn the causal relationships between attributes and affordances, we visualize the features of attributes and affordances in the image that have causal annotation relationships, with the results shown in Figure 9. Through the visualization, it can be observed that the regions of attributes and affordances with causal relationships are approximately similar. The visualization results indicate that specific attributes can be used to infer the corresponding affordances.

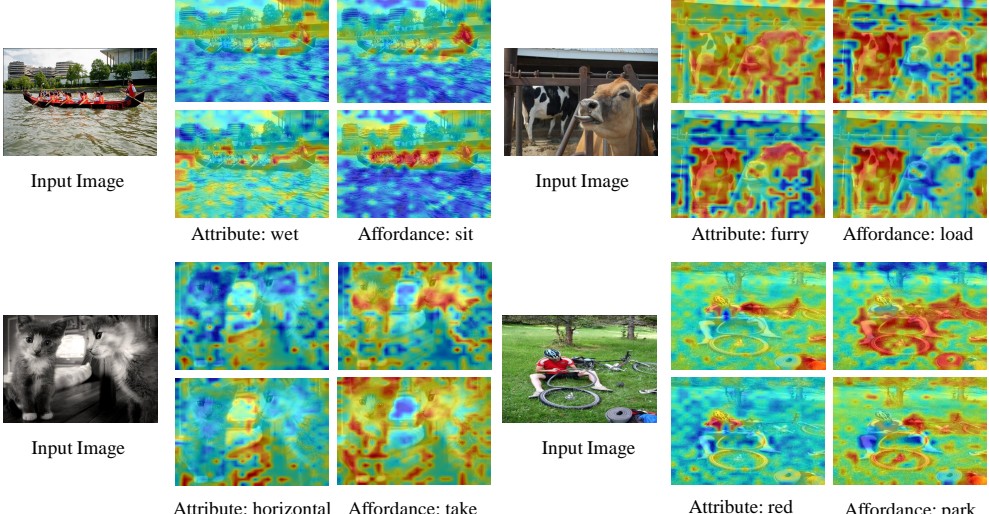

Figure 6: More heatmaps of Coarse-to-Fine Hierarchical Reasoning. For each image, the top side indicates the heatmaps from the coarse-grained prompt generation module, and the bottom side indicates the heatmaps from the fine-grained prompt formation module..

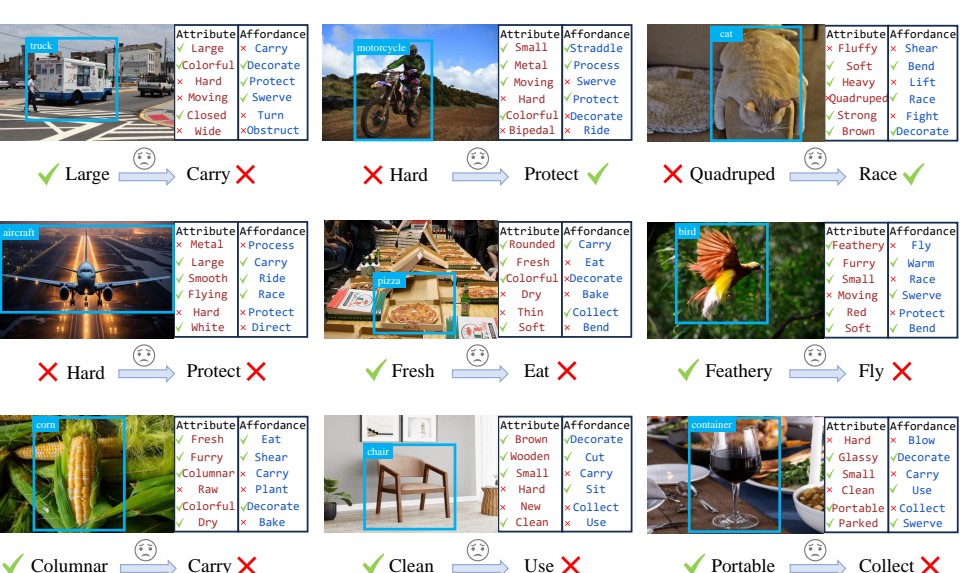

Figure 7: The prediction results of OCRN.

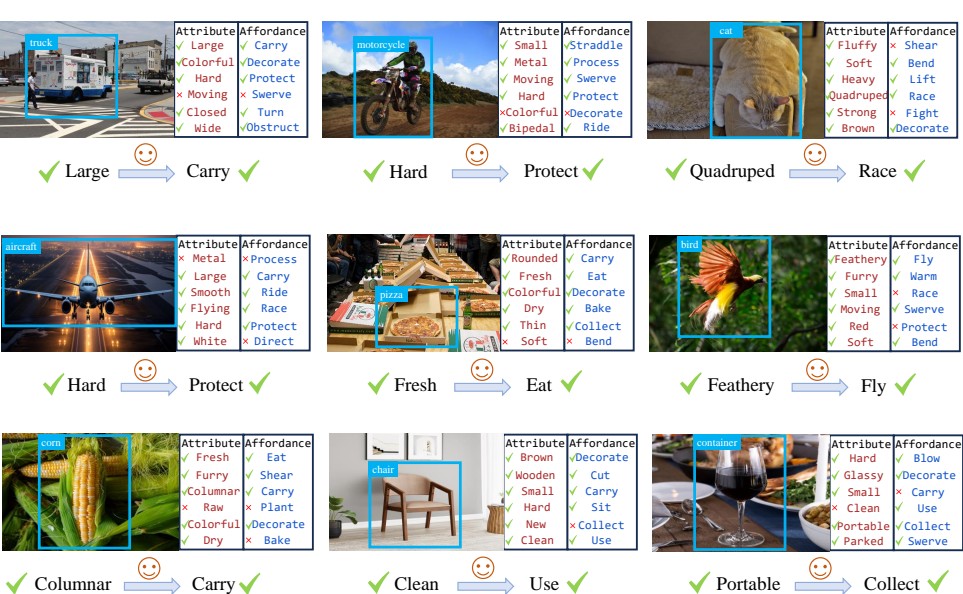

Figure 8: The prediction results of our HGR.

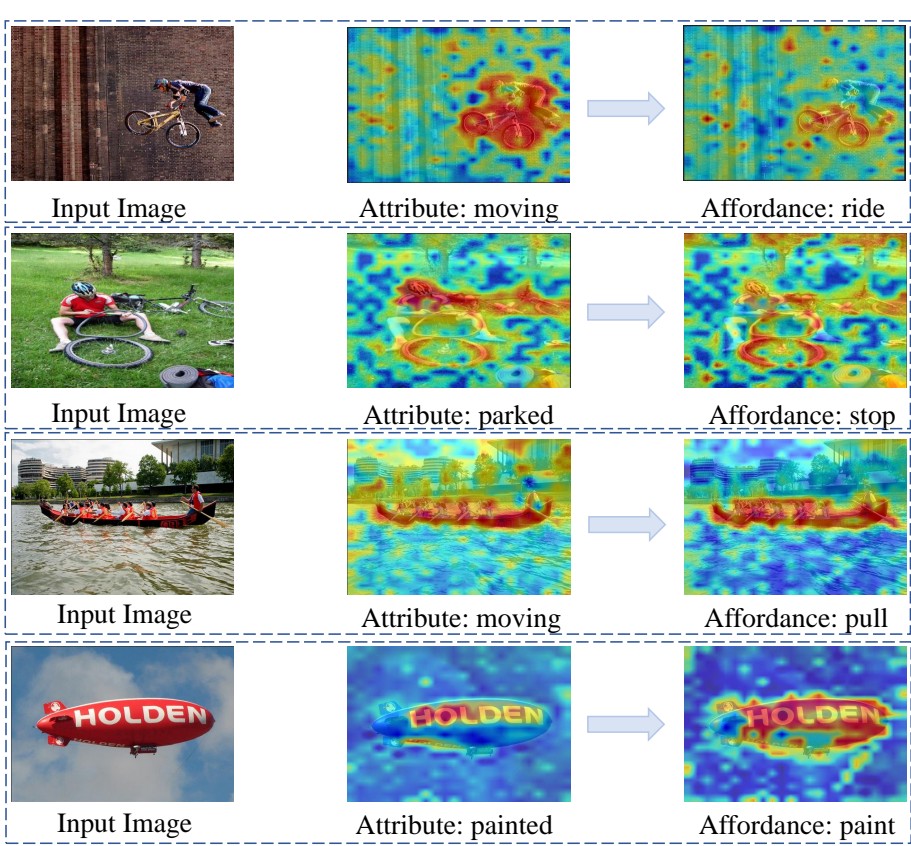

Figure 9: The visualization of causal relations.

