# OpenReview forum: "Hierarchical Multi-Grained Reasoning for Object Concept Learning"
_ICLR.cc/2025/Conference — Submitted to ICLR 2025_

### Official Review · Reviewer_c9Bi · 2024-10-28

**Soundness:** 3
**Presentation:** 3
**Contribution:** 3
**Rating:** 5
**Confidence:** 4

**Summary:**

The paper presents a promising new approach to OCL that leverages hierarchical reasoning and counterfactual samples to enhance model performance. The paper is well-written, and the results are compelling.

**Strengths:**

The paper introduces a novel Hierarchical Multi-Grained Reasoning (HGR) framework for Object Concept Learning (OCL), which is a significant step forward in addressing the many-to-many mapping challenge in OCL. The coarse-to-fine hierarchical reasoning module and the counterfactual relation-enhancing module are innovative components that show promise in improving reasoning accuracy.

**Weaknesses:**

1. The distributions of attribute and affordance concepts are usually imbalanced. For example, most clocks are round, and few clocks are square. A natural question is how the imbalance ratio influences the model performance. Does the proposed method work under extremely imbalanced cases?
2. The authors mainly compare the results with one work, which makes the results not very convincing.
3. The authors employ ground-truth bounding boxes to achieve fine-grained visual content learning. However, what should we do if these annotations are unavailable?
4. The authors mainly conduct experiments on natural images. The generalization to other domains (eg, medical data) is not clear. Also, the authors do not include a limitation section.

**Questions:**

See weaknesses above.

---

> ### Author Response · Authors · 2024-11-21
> **Author Response to Reviewer c9Bi**
>
> We thank reviewer c9Bi for the valuable and constructive comments. We address the concerns as follows.
>
> **Q1：How the imbalance ratio influences the model performance. Does the proposed method work under extremely imbalanced cases?**
>
> We agree with the reviewer that there is an imbalance in the distribution of attributes and affordances for objects. The original distribution of attributes and affordances in the OCL dataset is also imbalanced. To illustrate the impact of the imbalance ratio on our method, we set the same imbalance ratio $r$ for both attributes and affordance, and we conduct experiments as follows:
>
> *Table 1: Ablation of imbalance ratio $r$ on OCL. The values in "()" represent the imbalance ratio.*
> |  | **$α$** | **$β$** | **$S_{\mathbf{ITE}}$** | **$S_{\mathbf{α-β-ITE}}$** |
> |:-----:|:-----:|:-----:|:-------------------------:|:-------------------------:|
> |  $OCRN-r (100)$   | 26.4 | 49.8 |          15.5             |          14.3 |
> |  $HGR-r(100)$   | **29.5** | **52.0** |          **16.1**             |          **14.7** |
> |  $OCRN-r (50)$   | 29.1 | 52.3 |          18.2             |          16.1             |
> |  $HGR-r( 50)$   | **34.2** | **55.1** |          **18.9**            |          **16.6**             |
> |  $OCRN-r (10)$   | 32.4 | 54.1 |          20.4             |          16.7             |
> |  $HGR-r (10)$   | **39.2** | **57.2** |          **20.5**            |          **17.3**          |
>
> By setting the same imbalance ratio in the OCL baseline OCRN and our method HGR, it can be observed that our method still improves performance. This indicates that our approach can effectively work under extremely imbalanced cases.
>
> **Q2：The authors mainly compare the results with one work, which makes the results not very convincing.**
>
> To demonstrate the effectiveness of our method, we replace CLIP with the LLaVA-1.6 framework for validation, and the results are shown as:
>
> *Table 2: OCL accuracies with LLaVA-1.6 as baseline.*
> |  | **$α$** | **$β$** | **$S_{\mathbf{ITE}}$** | **$S_{\mathbf{α-β-ITE}}$** |
> |:-----:|:-----:|:-----:|:-------------------------:|:-------------------------:|
> |  **$ \text{LLaVA-1.6} $**   |  38.1 |  57.8 |          20.9            |          17.3             |
> |  **$ \text{LLaVA-1.6 + HGR} $**   |  **42.3** |  **60.1** |         **21.7**           |          **18.5**            |
>
> The results indicate that our method can enhance the existing vision-language models' ability to recognize object attributes and affordances. Furthermore, in Section 4.2, our method is applied to multi-task indoor scene understanding and weakly supervised affordance grounding tasks, and the accuracy is also improved significantly. This phenomenon further demonstrates the effectiveness of our approach.
>
> **Q3：What should we do if these annotations are unavailable?**
>
> In the OCL setting, the ground-truth bounding boxes are employed. We replace the bounding boxes with the predicted boxes extracted by Faster R-CNN. The results of our experiments are shown in Table 3. From the experimental results, it can be observed that there is a decline in performance after using the pre-trained Faster R-CNN to extract the prediction boxes. However, our method can still enhance the performance of the baseline.
>
> *Table 3: Ablation of ground-truth bounding boxes (bbox) on OCL.*
> |  | **$α$** | **$β$** | **$S_{\mathbf{ITE}}$** | **$S_{\mathbf{α-β-ITE}}$** |
> |:-----:|:-----:|:-----:|:-------------------------:|:-------------------------:|
> |  **$ \text{OCRN  w/ bbox } $**   |  31.5 |  53.6 |          20.3           |          16.9             |
> |  **$ \text{OCRN  w/o bbox } $**   |  28.6 |  49.8 |          17.7           |          15.5             |
> |  **$ \text{HGR w/o bbox } $**   |  **33.9** |  **53.3** |          20.1           |          **17.0**             |

---

> > ### Author Response · Authors · 2024-11-21
> > **Continue**
> >
> > **Q4: The generalization to other domains (eg, medical data) is not clear. Also, the authors do not include a limitation section.**
> >
> > Thanks for your suggestion. We have compared our method with the GLoRIA [ref1], BioViL [ref2] and MedKLIP [ref3] baselines. To ensure fairness, we follow the same protocol. The experiments are conducted on the ChestX-ray14  dataset.
> > From the experimental results, it can be observed that our method can achieve better performance enhancement, which demonstrates that our method possesses generalization ability. We will add the limitation section in the revised version.
> >
> > *Table 4: Comparison of AUC scores with other methods on fine-tuning classification task. The results are reported for ChestX-ray14 dataset.*
> > | Methods  |           |      Data portion 1%     |    Data portion 10%     |  Data portion 100%     |
> > |----------|-----------|:--------------:|:-----------:|:----------:|
> > | GLoRIA   |           |     0.6710    |    0.7642   |   0.8184   |
> > | BioViL   |           |     0.6952     |    0.7527   |   0.8245   |
> > | MedKLIP |           |     0.7721     |    0.7894   |   0.8323   |
> > | **HGR** |           |  **0.7876**    | **0.7963**  | **0.8388** |
> >
> > [ref1] Gloria: A multimodal global-local representation learning framework for label-efficient medical image recognition, ICCV 2021
> > \
> > [ref2] Making the most of text semantics to improve biomedical vision--language processing, ECCV 2022
> > \
> > [ref3] MedKLIP: Medical Knowledge Enhanced Language-Image Pre-Training in Radiology, arXiv

---

> ### Comment · Reviewer_c9Bi · 2024-11-26
>
> Thanks for the response. I have reviewed the comments from other reviewers, and the author's explanation partially addresses my concerns. I would maintain my rating.

---

> > ### Author Response · Authors · 2024-11-26
> > **Thanks for reading our rebuttal**
> >
> > Dear Reviewer,
> >
> > Thank you for reacting to our rebuttal.
> >
> > We are glad to hear that we have been able to address some of your concerns. However, it seems there are still a few points that could benefit from further clarification. We genuinely appreciate any additional suggestions you may have to help us refine and improve our work.
> >
> > Thank you again for taking the time to read our rebuttal and for providing your review.

---

### Official Review · Reviewer_3E1s · 2024-11-02

**Soundness:** 2
**Presentation:** 2
**Contribution:** 3
**Rating:** 5
**Confidence:** 4

**Summary:**

This paper designs an object concept learning method leveraging hierarchical and counterfactual structure to achieve better many-to-many mapping in the OCL problem. The proposed method first mimics the human coarse-to-fine reasoning process to learn category-independent attributes and affordances, then introduces counterfactual supervision to strengthen relationships between them. The method comprises four modules: Coarse-grained Prompt Generation, Fine-grained Prompt Formation, Visual Concept Extraction, and Concept Connectivity with Counterfactual, realizing coarse-to-fine hierarchical reasoning and counterfactual relation-enhancing respectively. The paper conducts experiments on multiple tasks, including Object Concept Learning, Multi-task Indoor Scene Understanding, and Weakly Supervised Affordance Grounding. The results show that the proposed method has a certain performance advantage compared to the SOTAs.

**Strengths:**

1. The paper observes the relationship between attributes and affordances, and proposes a reasoning framework between them to solve the OCL problem. Experiments demonstrate that reasoning between the two indeed aids in the recognition of the two types of concepts.
2. The paper contains a lot of visualization contents for intermediate results and experiments, making the results intuitive and easy to understand.

**Weaknesses:**

1. Multi-label recognition is not a new problem. It does not demonstrate how the proposed method addresses the many-to-many problem.
2. The proposed method utilizes a large amount of supervised information, including ground-truth bounding boxes, as well as causality annotations between attributes and affordances. This poses a challenge to the generalizability of the method.
3. Regarding the problem definition: How are C_\alpha and C_\beta selected in Section 3.2.1? Are \alpha and \beta here consistent with h_\alpha and h_\beta in Eq.1? A more accurate and detailed supplementary explanation is needed here.
4. The expression for concept distinctiveness loss in Eq. 9 is problematic; the summation expression does not include j. This should be more rigorous.
5. Some minor issues: In Figure 1, "clolrful" seems to be a typo for "colorful," and there is an incorrect duplicate of k_alpha in Line 470.

**Questions:**

1. Please clarify the difference between object concept learning and multi-label object recognition. Is it feasible to treat concepts directly as categories?
2. Does choosing the same k for attributes and affordances affect the generalizability of the method? If k concepts are sampled from the complete dataset, is the model capable of recognizing all concepts?
3. What is the necessity of using GRU and concept update in Visual Concept Extraction?
4. It is recommended to place the names of the SOTA methods in Sec. A.1 within the main text. Otherwise, it's difficult to associate the methods in the Table 1 with the references.

---

> ### Author Response · Authors · 2024-11-21
> **Author Response to Reviewer 3E1s**
>
> We thank Reviewer 3E1s for the valuable comments.
>
> **Q1: It does not demonstrate how the proposed method addresses the many-to-many problem.**
>
> Although multi-label recognition is not a new problem, it is under explored in the Object Concept Learning (OCL) task. Multi-label recognition mainly focus on the mapping between images and categories. The main challenge of OCL lies in the many-to-many mapping relationships between objects and concepts. That is, an object could have multiple different attributes and affordances (concepts). Meanwhile, an attribute and affordance could also belong to multiple different objects. To overcome this, we propose a Hierarchical Multi-Grained Reasoning framework. **For objects in an image, an attribute and affordance could belong to multiple different objects, which means concepts exist across objects. Thus, we first design object-agnostic prompts to enhance concepts-objects mapping precision. Subsequently, a mapping between objects-concepts is formed by integrating contextual global information and fine-grained local information.** Furthermore, to make the mapping between objects and concepts more precise, we introduce counterfactual reasoning to identify causal relationships between certain attributes and affordances. This enhances the mapping connections between objects and these concepts, which in turn improves recognition performance. Significant performance improvements and extensive visual analysis have demonstrated the superiority of HGR.
>
> **Q2: The generalizability of our method.**
>
> To ensure a fair comparison with existing methods, we follow the same settings of the OCL [ref1] task, utilizing ground-truth bounding box and causality annotations. The experimental results in Section 4 demonstrate the effectiveness of our approach.
> Furthermore, to validate the generalizability of our proposed method, we present our method application in Multi-task Indoor Scene Understanding and Weakly Supervised Affordance Grounding tasks in Section 4.2. For fairness, we maintain the same settings as the baselines in these tasks, without grounding-truth bounding box and causality annotations. In Weakly Supervised Affordance Grounding task, our method outperforms the SOTA method by 5.2%. In the challenged Multi-task Indoor Scene Understanding task, our method outperforms the SOTA method by 0.9%. In addition, we also evaluate our method in the medical dataset, which can be seen in Reviewer c9Bi Q4. The experimental results show that our method could enhance the prediction accuracy, highlighting the generalizability of our approach.
>
> [ref1] Beyond Object Recognition: A New Benchmark towards Object Concept Learning, ICCV 2023
>
> **Q3: Explanation of problem definition.**
>
> The values of $C_{\alpha}$ and $C_{\beta}$ are initialized using the visual features $F_{a}$ . Through attention-based aggregation, they are progressively updated. Here, $\alpha$ and $\beta$ represent attributes and affordances, respectively, which correspond to the subscripts $h_{\alpha}$ and $h_{\beta}$ in Equation (1).
>
> **Q4：Expression issue.**
>
> Thank you for pointing out the expression issues. We will make corrections in the revised version.
>
> **Q5: Please clarify the difference between object concept learning and multi-label object recognition. Is it feasible to treat concepts directly as categories?**
>
> Object Concept Learning (OCL) task aims to push the envelope of object understanding rather than just recognizes object categories. It requires machines to reason out attributes and affordances, and simultaneously give the reason: what attributes make an object possess these affordances [ref1].
>
> Attributes and affordances should not be directly treated as categories. Category apple is a symbol indicating its referent (real apples). Attributes depict object states [ref2]. An elegant characteristic of attributes is cross-category: objects of the same category can have various states (big or fresh apple), while various objects can have the same state (sliced orange or apple). Affordance indicates what actions humans can perform with objects, and it is also cross-category.
> If the category is the first level of object concept, the attribute can be seen as the second level closer to the physical fact. And affordance can be seen as the third level, which is closely related to common sense and causal inference [ref3].
>
> [ref1] Beyond Object Recognition: A New Benchmark towards Object Concept Learning, ICCV 2023
> \
> [ref2] Discovering states and transformations in image collections, CVPR 2015
> \
> [ref3] The ecological approach to the visual perception of pictures. Leonardo 1978

---

> > ### Author Response · Authors · 2024-11-21
> > **Continue**
> >
> > **Q6: Does choosing the same k for attributes and affordances affect the generalizability of the method? If k concepts are sampled from the complete dataset, is the model capable of recognizing all concepts?**
> >
> > Q6-1: In Section 4.3, we discuss the impact of changing the number of $k$ on OCL dataset. The conclusion is that the best recognition performance is achieved when the values of $k$ are the same. For other dataset like NYUd2 in multi-task indoor scene understanding task, the number of $k$ is different. We will make a particular explanation in the revised version.
> >
> > Q6-2:
> > We conduct experiments where the number of k samples are equal to the total number of attributes (114) and affordances (170) in the entire dataset, and the results are as follows:
> >
> > *Table 1: Ablation analysis for concept number $k$. "att" is the abbreviation for attributes, and "aff" is the abbreviation for affordances. The values in "()" represent the number of $k$.*
> > |  | **$α$** | **$β$** | **$S_{\mathbf{ITE}}$** | **$S_{\mathbf{α-β-ITE}}$** |
> > |:-----:|:-----:|:-----:|:-------------------------:|:-------------------------:|
> > |  att(114) - aff(114)   | 36.7 | 55.9 |          19.9             |          16.3
> > |
> > |  att(114) - aff(170)   | 35.9 | 55.1 |          19.5             |          15.9             |
> > |  att(10) - aff(10)   | 39.6 | 57.5 |          20.9             |          17.4             |
> >
> > The experimental results reveal that an excessive number of k samples leads to a decrease in performance, and a significant difference between attributes and affordances also results in poor performance. This is because learning too many visual concepts increases the complexity of the network and introduces information redundancy, which in turn degrades performance.
> >
> > **Q7: What is the necessity of using GRU and concept update in Visual Concept Extraction?**
> >
> > From the Table 2, it can be observed that removing the GRU leads to a slight decrease in performance. This is because the GRU is used to refine the shape of the regions corresponding to the extracted concepts. Different attributes and affordances of an object correspond to different region. Therefore, it is necessary to provide a better regional boundary for these concepts.
> >
> > *Table 2: Ablation of GRU on OCL.*
> > |  | **$α$** | **$β$** | **$S_{\mathbf{ITE}}$** | **$S_{\mathbf{α-β-ITE}}$** |
> > |:-----:|:-----:|:-----:|:-------------------------:|:-------------------------:|
> > |  **$ \text{w/o GRU} $**   |  39.4 |  57.2 |          20.7            |          17.3             |
> > |  **$ \text{w/ GRU} $**   |  39.6 |  57.5 |          20.9           |          17.4            |
> >
> > The experimental results in Table 3 indicate that performance declines when concept update is not employed. This is because each concept feature is initialized by visual features and then progressively updated to different object regions through attention-based clustering. Without concept updates, different concepts may intertwine, making it more difficult to complete the recognition task.
> >
> > *Table 3: Ablation of concept update on OCL.*
> > |  | **$α$** | **$β$** | **$S_{\mathbf{ITE}}$** | **$S_{\mathbf{α-β-ITE}}$** |
> > |:-----:|:-----:|:-----:|:-------------------------:|:-------------------------:|
> > |  **$ \text{w/o concept update} $**   |  38.8 |  57.0 |          20.5            |          17.1             |
> > |  **$ \text{w/ concept update } $**   |  39.6 |  57.5 |          20.9           |          17.4            |
> >
> > **Q8：It is recommended to place the names of the SOTA methods in Sec. A.1 within the main text.**
> >
> > Thanks for your suggestion. We will make the adjustments in the revised version.

---

> > > ### Comment · Reviewer_3E1s · 2024-11-28
> > >
> > > I thank the authors for their response by conducting further experiments and analyses. The answers addressed some of my concerns, however, I still think the novelty and technical contribution is limited and have concerns on its practical usage in real-world scenario. Therefore, I will maintain my score.

---

### Official Review · Reviewer_AyMp · 2024-11-02

**Soundness:** 3
**Presentation:** 3
**Contribution:** 2
**Rating:** 5
**Confidence:** 4

**Summary:**

This paper focus onobject concept learning (OCL) task and introduces a multi-component model to extract attribute and affordance concepts while do causal reasoning among them. The model employs coarse and fine-grained modules to capture global and local (within the GT box) features, which are then fed into a slot-attention-based concept extractor to predict attributes and affordances. A GNN is then utilized to learn the causal relationships between attributes and affordances. The method is evaluated across various benchmarks, showing superior performance.

**Strengths:**

- The writting is clear, making the method very understandable.
- The proposed model and its implementation are robust and self-contained.
- I would appreciate that the authors conducted extensive experiments on multiple benchmarks and the ablation study is thorough. The proposed method also significently outperforms the baselines.

**Weaknesses:**

1. the proposed method appears to be a modern implementation of OCL baselines. The Coarse-to-Fine Hierarchical Reasoning is the global and local feature embedders implemented in CLIP-style. The Visual Concept Extraction is the attribute/affordance classifier implemented with slot attention. And the causal inference module is purely a GNN. While all components are straightforwardly implemented, this may diminish the novelty of the method.
2. The causal component is implemented by an GNN supervised by causal annotations. This trivialize the causal inference part to a supervised learning task. The apporach primarily learns correlations between annotated labels thus would be doubted in the field of causality. And the partial annotations in OCL also leads to biased supervised learning of causal relations. As OCL suggests that causal annotations are primarily built for evaluation rather than training, a "real" causal inference module is essential in this context. (As we can see, HGR is not evaluated on zero-shot causal inference task).
3. The model size seems considerably larger than that of the baselines. A comparison would be appreciated.
4. Additional baselines is needed for comparison, as the vanilla CLIP alone is too simple. And It would be beneficial to compare against the multimodal LLMs (e.g. GPT4o). However, given the time and budget constraints, this may not factor into my final assessment.

**Questions:**

Major questions have been listed in the Weaknesses part.

Minor typos: Figure 1: "clolful". And "[P][T][P]" does not seem to correspond with the equation 1 below.

---

> ### Author Response · Authors · 2024-11-22
> **Author Response to Reviewer AyMp**
>
> We thank reviewer  AyMp for the valuable and constructive comments. We address the concerns as follows.
>
> **Q1：The novelty of the method.**
>
> Our main contribution is not another implementation of the OCL baseline, but proposing a Hierarchical Multi-Grained Reasoning framework. This framework leverages vision-language model to progressively guide the mapping of many-to-many relations between objects and concepts (object-concepts and concept-objects). Additionally, it introduces counterfactual reasoning to refine the mapping relations further, enabling the model not only to identify attributes and affordances but also to understand the causal relationships between them.
> Current methods for improving the accuracy of many-to-many mappings primarily focus on enhancing the discriminative features of objects.
> In this work, we first explore designing a dedicated reasoning method for learning object concepts, achieving promising results.
>
> **Q2: Explanation of causal relations.**
>
> We first clarify that we construct the counterfactual reasoning process during training. By imposing interventions on attribute prompts that have attribute-affordance causal relationships and designing counterfactual losses, the model could learn causal relationships.  The improvement in causality metrics $S_{\mathbf{ITE}}$ and $S_{\mathbf{α-β-ITE}}$ prove the effectiveness of our method.
> To further demonstrate the effectiveness of our method, we conducted experiments on the zero-shot causal setting, and the results are as follows:
>
> *Table 1: Performance results for OCL dataset on zero-shot causal learning task.*
> |  | **$α$** | **$β$** | **$S_{\mathbf{ITE}}$** | **$S_{\mathbf{α-β-ITE}}$** |
> |:-----:|:-----:|:-----:|:-------------------------:|:-------------------------:|
> |  **$ \text{OCRN} $**   |  30.0 |  52.5 |          16.3            |          14.1             |
> |  **$ \text{HGR} $**   |  **37.5** |  **56.1** |          **17.3**           |          **15.2**            |
>
> The results indicate that our method can enhance the model's reasoning performance. In this setting, 300 attributes-affordances causality annotations are used as unseen causal relations.  785 attributes-affordances causality annotations are used as seen causal relations. More details will be updated in the revised version.
>
>
> **Q3: The model size seems considerably larger than that of the baselines. A comparison would be appreciated.**
>
> Thanks for your suggestions. The model size is indeed larger than other baselines. We compare our method HGR with the baselines, and the results are as follows:
>
> *Table 2: Statistics of Param.*
> | **Model** | **#Param.** |
> |:-----:|:-----:|
> |  HGR  |  971M |
> |  Vanilla CLIP  |  563.12M |
> |  OCRN  |  143.25M |
>
> **Q4: Additional baseline comparision.**
>
> Thanks for your suggestions. We have made our best efforts to conduct experiments on LLaVA-1.6 as additional baseline, and the results are as follows:
>
> *Table 3: OCL accuracies with LLaVA-1.6 as baseline.*
> |  | **$α$** | **$β$** | **$S_{\mathbf{ITE}}$** | **$S_{\mathbf{α-β-ITE}}$** |
> |:-----:|:-----:|:-----:|:-------------------------:|:-------------------------:|
> |  **$ \text{LLaVA-1.6} $**   |  38.1 |  57.8 |          20.9            |          17.3             |
> |  **$ \text{LLaVA-1.6 + HGR} $**   |  **42.3** |  **60.1** |          **21.7**           |          **18.5**            |
>
> The results indicate that our method can enhance the existing vision-language models' ability to recognize object attributes and affordances.
>
> **Q5: Minor typos**
>
> Thank you for pointing out the relevant issues. We will make corrections in the revised version.

---

> > ### Comment · Reviewer_AyMp · 2024-11-25
> >
> > Thanks. I have read the revised draft and the authors' and other reviewers' comments and some of my concerns are addressed.
> >
> > **Q1-Q2** : Thanks for the clarification. However, the causal relation could not be simply learned by this supervised learning paradigm, as in most causal inference cases, there are no supervision data.
> >
> > **Q3-Q4** : Thanks for the additional tabulars. Especially, the comparison to MLLM could make the experiments more solid and robust.
> >
> > Overall, the paper is solid in empirical experiments, but the motivation slightly lacks. I would keep my rating.

---

> > > ### Author Response · Authors · 2024-11-26
> > > **Further Response to Reviewer AyMp**
> > >
> > > We sincerely appreciate the reviewer’s feedback. We value your feedback and would like to further provide a detailed response to address the points you raised.
> > >
> > > **Q1: However, the causal relation could not be simply learned by this supervised learning paradigm, as in most causal inference cases, there are no supervision data.**
> > >
> > > We agree that most causal inference cases have no supervision data. However, as stated in [ref1], one of the goals of OCL is to accurately learn these causal relations among affordance and attributes using supervised paradigm. Moreover, as presented in Rebuttal Table 1, zero-shot experiments demonstrate that our method can generalize to the unseen causal relations 2.8% using supervision from seen causal relations 7.2%, further validating the effectiveness of our approach.
> > >
> > > We sincerely appreciate the reviewer’s constructive suggestions and believe that the additional experiments, analysis, and explanations significantly improve the quality of our submission.
> > >
> > > [ref1] Beyond Object Recognition: A New Benchmark towards Object Concept Learning, ICCV 2023

---

### Official Review · Reviewer_U8nc · 2024-11-03

**Soundness:** 2
**Presentation:** 2
**Contribution:** 2
**Rating:** 5
**Confidence:** 2

**Summary:**

This submission proposes Hierarchical Multi-Grained Reasoning approach for object concept learning. They first presents a coarse-grained prompt generation strategy to enhance attribute and affordance description, and then incorporates augmented samples as a reasoning approach to obtain fine-grained representations. Extensive experiments demonstrate the effectiveness of the proposed method.

**Strengths:**

1. The paper is well organized, and easy to understand.
2. The experiments demonstrate the effectiveness of the proposed method.
3. Multi-grained representations and fine-grained representations for object concept learning look like reasonable.

**Weaknesses:**

1. I am confusing about the reasoning description in Section 3.2. The discussion of this paper majorly relies on that attribute and affordances are causal relationship, which slightly confuse me. Let's pick up the example in Fig. 5. Do you think "Furry" is the cause of the affordance "shear", "Small" is the cause of affordance "Carry"? I'd like to admit that the definition of attribute is ambiguous. Maybe you can find some attribute is the cause of a particular affordance. However, a lot attributes of the object are not causally related to affordance. From this point, the motivation of the core part in this paper is questionable.
2. Multi-grained representation and augmented representation for robust representation are common in deep neural networks.
3. Assuming the neural network might implicitly utilize the possible causal relationship behind the affordance, while the visualized illustration indicates the selected attributes are not the reason of corresponding affordance. The paper neither demonstrates the causal relations.
4. I think the causality behind affordance is interesting. However, this paper does not demonstrate convincible causal relation. From my point, it is merely correlation.

**Questions:**

I have a questions, do you think current MLLM model can easily address affordance and attributes recognition?

Please also refer to the weaknesses.

---

> ### Author Response · Authors · 2024-11-22
> **Author Response to Reviewer U8nc**
>
> We thank reviewer U8nc for the valuable and constructive comments. We address the concerns as follows.
>
> **Q1: The causal relationships between attributes and affordances.**
>
> We agree with the reviewer's viewpoint that not all attributes of the object have causal relationships with affordances. In the OCL setting and the OCL dataset construction, this problem has been considered and solved by choosing about 10% convinced attribute-affordance classes as attribute-affordance causal pair candidates [ref1]. Thus, as described in Section 3.2.2, we design counterfactual reasoning based on the causality annotation. Additionally, the causality pairs shown in Figure 5 exist in the causal annotations of the dataset. To address these concerns and clarify this point further, we will clarify it in the revised version.
>
> [ref1] Beyond Object Recognition: A New Benchmark towards Object Concept Learning, ICCV 2023
>
> **Q2: Multi-grained representation and augmented representation for robust representation are common in deep neural networks.**
>
> While multi-grained and augmented representations are common in deep neural networks, these approaches have yet to be well explored for OCL tasks.  The OCL task not only identifies the higher-level knowledge attributes and affordances of objects but also reasons out the causal relationships between specific attributes and affordances. **Our method is not merely a simple multi-grained feature augment approach; instead, we design a hierarchical reasoning framework that integrates contextual information and causality to reason out higher-level knowledge attributes and affordances.** Moreover, since causal relationships exist between specific attributes and affordances, we introduce a counterfactual reasoning module to help the model more precisely capture these causalities during training, thereby improving concepts' recognition performance. This is not just a simple feature augmentation but a facilitation for the model to infer the causal relationships behind affordances. Experimental results validate the effectiveness of our method.
>
> **Q3：The demonstration of causal relations.**
>
> We adopt causal metrics $S_{\mathbf{ITE}}$ and $S_{\mathbf{α-β-ITE}}$ for evaluation in experiment part Table 1 to demonstrate the extent to which the model learned causal relationships. $S_{\mathbf{ITE}}$ evaluates the model's causal reasoning ability by measuring the attribute-affordance causality prediction results.
> $S_{\mathbf{α-β-ITE}}$ means multiplying $S_{\mathbf{ITE}}$ with predicted probabilities.
> It can be seen that our method has achieved improvements on both of these metrics.
>
> Additionally, to qualitatively show the causal relation between specific attributes and affordances, we will upload our visualization results in Appendix A.8 of the revised version.
> Particularly, our visualizations in Figure 3 and Figure 6 just aim to show that the model could progressively focus on attribute and affordance features through hierarchical reasoning.
>
> **Q4: Do you think current MLLM model can easily address affordance and attributes recognition?**
>
> We think that MLLM has the potential to address the problem of attribute and affordance recognition, but MLLM still faces some challenges. For instance, in [ref1], [ref2], and [ref3], MLLM is used as an external knowledge base to provide some attribute (small, long) and affordance (hold, grasp) knowledge.
> However, classic MLLM is generally trained on generic image-text pairs, lacking critical robotic knowledge to understand object affordances and their physical properties [ref4] [ref5]. For example, the machine needs to understand objects' geometric structure to predict the movable contact regions.
> Enabling MLLM to solve the recognition of attributes and affordances easily still requires sustained attention and research.
>
> [ref1] ImplicitAVE: An Open-Source Dataset and Multimodal LLMs Benchmark for Implicit Attribute Value Extraction, arxiv 2024
> \
> [ref2] INTRA: Interaction Relationship-aware Weakly Supervised Affordance Grounding, ECCV 2024
> \
> [ref3] WorldAfford: Affordance Grounding based on Natural Language Instructions, arxiv 2024
> \
> [ref4] ManipVQA: Injecting Robotic Affordance and Physically Grounded Information into Multi-Modal Large Language Models, arxiv 2024
> \
> [ref5] ManipLLM: Embodied Multimodal Large Language Model for Object-Centric Robotic Manipulation, CVPR 2024

---

> > ### Comment · Reviewer_U8nc · 2024-11-25
> >
> > Thanks for your response.
> >
> > I think my core concern is not addressed. As shown in Q1, there are only 10% possible causal relations among affordance and attributes. Sometimes, we categorized some kinds of affordance into attributes in some literature. Therefore, it is unconvincing that it is the causality of affordance and attributes. I  agree with the point of AyMp. I think there are correlations rather than causality.

---

> > > ### Author Response · Authors · 2024-11-26
> > > **Further Response to Reviewer U8nc**
> > >
> > > Thank you for highlighting your core concern. We value your feedback and would like to provide a detailed response to address the points you raised.
> > >
> > > **Q1-1: There are only 10% possible causal relations among affordance and attributes. Therefore, it is unconvincing that it is the causality of affordance and attributes.**
> > >
> > > We clarify that, as stated in [ref1], one of the goals of OCL is to accurately learn these 10% causal relations among affordance and attributes using supervised paradigm. To ensure that these 10% samples are indeed causal relations, in the OCL setting, human experts are asked to determine and check the causal relationship that exists between attributes and affordances. The majority votes are taken and the not-sure and controversial pairs are rechecked. The not-sure and no pairs are removed, and so do the ambiguous pairs [ref1].
> > >
> > > Moreover, as presented in Table 1, zero-shot experiments demonstrate that our method can generalize to the unseen causal relations 2.8%  using supervision from seen causal relations 7.2%, further validating the effectiveness of our approach.
> > >
> > > *Table 1: Performance results for OCL dataset on zero-shot causal learning task.*
> > > |  | **$α$** | **$β$** | **$S_{\mathbf{ITE}}$** | **$S_{\mathbf{α-β-ITE}}$** |
> > > |:-----:|:-----:|:-----:|:-------------------------:|:-------------------------:|
> > > |  **$ \text{OCRN} $**   |  30.0 |  52.5 |          16.3            |          14.1             |
> > > |  **$ \text{HGR} $**   |  **37.5** |  **56.1** |          **17.3**           |          **15.2**            |
> > >
> > > In this setting, 300 attributes-affordances causality annotations are used as unseen causal relations.  785 attributes-affordances causality annotations are used as seen causal relations.
> > >
> > > [ref1] Beyond Object Recognition: A New Benchmark towards Object Concept Learning, ICCV 2023
> > >
> > > **Q1-2: I think there are correlations rather than causality.**
> > >
> > > We first clarify that, in the OCL setting, we focus exclusively on attribute-affordance pairs with explicit causality, as described in the OCL paper [ref1]: *"It should be mentioned that we only annotate whether an attribute-affordance pair has explicit and key causality."*
> > >
> > > Admittedly, among all attribute and affordance, some pairs exhibit statistical correlations. For example, the fresh apples on a shelf and the apples' peel-able are correlated. However, there is also convincing, true causality between attributes and affordances, such as “fresh” causes “eat-able”. **We only focus on attribute-affordance pairs with true causal relationships from the human expert checked as mentioned in Q1-1.**
> > >
> > > Additionally, we construct counterfactual samples based on real causal pairs. For instance, if an apple is not fresh, the model should predict it as inedible rather than unpeelable. This approach avoids interference from no causality attribute-affordance pairs, facilitating the learning of precise causal relationships.
> > >
> > > Finally, we would like to emphasize that we enhance the model's ability to recognize attributes and affordances of objects by designing a hierarchical multi-grained reasoning framework to capture the many-to-many mapping relationships more accurately. Extensive experimental results and visualizations strongly validate our method.
> > >
> > >
> > > We sincerely appreciate the reviewer’s constructive suggestions and believe that the additional experiments, analysis, and explanations significantly improve the quality of our submission. We hope that this provides a sufficient reason to consider raising the score.

---

> ### Comment · Reviewer_U8nc · 2024-11-26
>
> Thanks for your response and the provided references. The annotated causal relations are questionable. I think it relies on a strong assumption which the paper did not claim. By the way, ``fresh'' does not cause ``eatable'', even if it is limited to the figures in the paper. For the use of these keywords, concepts, and claims, I'd like to refer to Judea Pearl's comments about the recent use of those concepts. Though I think it is confusing about the causality, I keep boardline reject given that this paper has done extensive experiments,

---

> > ### Author Response · Authors · 2024-12-02
> > **Further Response to Reviewer U8nc**
> >
> > We sincerely appreciate your valuable feedback, which has been instrumental in improving our submission.
> >
> > **Q1-1：The annotated causal relations are questionable. I think it relies on a strong assumption which the paper did not claim.**
> >
> > In L292 of our paper, we point out that we follow the OCL [ref1] setup and use the causal annotations from the OCL benchmark. Additionally, many works such as [ref2], [ref3], and [ref4] also aim to combine supervised deep learning with causal inference. We will provide further details about the causal annotations in future version.
> >
> > **Q1-2：By the way, fresh'' does not cause eatable'', even if it is limited to the figures in the paper.**
> >
> > We do not fully agree with this viewpoint. We clarify that for a specific object, such as an apple, we often judge its "eatable" by observing its attributes, such as "fresh", "clean", and "whole".
> > If "fresh" is not the direct cause of "eatable",  it is still reasonable to consider it as a condition that enables the cause to yield its effect, i.e., enabling condition.
> >  In modern causal reasoning models, such as Structural Causal Models, there is no strict distinction between causal relationships and enabling conditions. As Cheng et al. [ref5] state, causes and enabling conditions hold the same logical relation to the effect in those terms, and the methods that explain their distinction come from the subject judgment of humans. As stated in OCL [ref1], OCL follows the “open” setting: affordance is a subjective property of the object, so all reasons given by humans/robots (including enabling conditions) are regarded as causal factors.
> >
> > **Q1-3: For the use of these keywords, concepts, and claims, I'd like to refer to Judea Pearl's comments about the recent use of those concepts.**
> >
> > Judea Pearl has mentioned in _<<The Book of Why>>_  that "**the ability to reason about the intentions (i.e., consciousness)** has always been a major challenge for researchers in the field of artificial intelligence, and this ability is also what defines the concept of an "agent". I believe that **the algorithmization of counterfactuals is an important step** in transforming consciousness and intelligent agents into computational reality."
> >
> > Similarly, in the OCL task, we expect to accurately identify attributes and affordances from objects, and **the counterfactual relation-enhancing model in our method** is employed to capture causal relationships across different instances, attributes and affordances, thereby **enhancing the model's ability to reason out object affordances** and pushing the envelope of object understanding.
> >
> > [ref1] Beyond Object Recognition: A New Benchmark towards Object Concept Learning, ICCV 2023
> > \
> > [ref2] Discovering causal signals in images, CVPR 2017
> > \
> > [ref3] ML4S: Learning Causal Skeleton from Vicinal Graphs, KDD 2022
> > \
> > [ref4] ML4C: Seeing Causality Through Latent Vicinity, ICDM 2023
> > \
> > [ref5] Causes versus enabling conditions, Cognition 1991
> >
> >
> > We hope the above clarification addresses your concern.We thank you for your patience and support.

---

### Meta-Review · Area_Chair_SLCR · 2024-12-15

**Metareview:**

The reviewers unanimously rated the paper negatively. They appreciated the clear presentation, strong performance, and extensive experiments of the paper. However, they also raised concerns with the proposed approach to learning causal relations (U8nc, AyMp), lack of technical novelty (U8nc, AyMp, 3E1s), high space-time complexity (AyMp), limited baselines to be compared (AyMp, c9Bi), relying on a large amount of supervision which may lead to a generalization issue (3E1s, c9Bi), and a few presentation issues (3E1s).

The authors' rebuttal and subsequent responses in the discussion period address some of these concerns but failed to fully assuage all of them, e.g., those about causal relation learning and limited technical contribution. In particular, the reviewers were still unsure about the validity of the proposed causal learning strategy: they argued that the supervised learning with partial causal annotations is highly likely to learn correlations rather than causal relations and to lead to a biased model (U8nc, AyMp). Also, Reviewer AyMp pointed out that the causal annotations in OCL are given for testing, not training, which was confirmed by AC's reading of the OCL benchmark paper. Unfortunately, the last response of the authors did not fully resolve these concerns as the experimental results reported in the response do not come along with a proper justification.

Putting these together, the AC found that the remaining concerns outweigh the positive comments and the rebuttal, and thus regrets to recommend rejection. The authors are encouraged to revise the paper with the comments by the reviewers and the AC, and submit to an upcoming conference.

**Additional Comments On Reviewer Discussion:**

- **The proposed strategy for learning causal relations (U8nc, AyMp)**: The AC weighed this issue heavily when making the final decision. Some of the original comments of Reviewer U8nc on this issue, e.g., attributes of an object unnecessarily related causally to affordance and lack of prope empirical demonstration, have been successfully addressed by providing details of the OCL benchmark dataset. However, the other concerns have not resolved appropriately. Reviewer AyMp argued that the supervised learning using partial causal annotations of the dataset will lead to a model that is biased and learns correlations instead of causal relations; Reviewer U8nc agreed with this. The authors' response to this comment was a simple statement without a proper logical or theoretical justification, and the experimental results were not convincing, according to Reviewer AyMp. Further, Reviewer AyMp mentioned that causal annotations in OCL are actually given for testing, not training, which seems to be true according to Section 4.2 of the benchmark paper. Hence, the AC suspects that this is an abuse of the causal annotations and the submission could lead the community down the wrong path if it is accepted as is.
- **Lack of technical novelty (U8nc, AyMp, 3E1s)**: The reviewers considered the proposed method as a combination of existing techniques, and the authors argued that, although each component has been introduced in the literature, the combination is novel in OCL and the proposed hierarchical multi-grained approach is a novel direction to OCL. This argument unfortunately failed to fully assuage all the three reviewers; one of them still mentioned that this is the major reason for rejection. However, the AC did not weigh this issue heavily when making the final decision as the authors' argument sounds reasonable.
- **High space-time complexity (AyMp)**: The authors provide an additional analysis result, which reveals that this concern is true. However, this is not a great issue especially when considering the OCL task has not been studied enough.
- **Limited baselines to be compared (AyMp, c9Bi)**: This concern has been well resolved by providing additional experiments.
- **Relying on a large amount of supervision which may lead to a generalization issue (3E1s, c9Bi)**: The AC found that this concern has been resolved by providing details of the OCL benchmark.
- **A few presentation issues (3E1s)**: These are minor issues that did not affect the final decision.

---

### Decision · Program_Chairs · 2025-01-22

Reject